# Assessing the drivers of sexual behavior among youth and its social determinants in Nepal

**Grishu Shrestha**[ID]**, Reecha Piya, Sampurna Kakchapati**[ID]*****, Parash Mani Sapkota**[ID]**,
Deepak Joshi, Sushil Chandra Baral**

Health Research and Social Development Forum (HERD) International, Lalitpur, Nepal

* kck_sampurna@yahoo.com, sampurna.kakchapati@herdint.com

## Abstract

### Introduction

Sexual behavior among youth is a public health concern, particularly in contexts where cultural norms, socio-economic factors, and access to comprehensive sexual education play pivotal roles. This paper aims to examine the determinants of sexual behavior among Nepali youths.

### Methods

This study analyzed data from 7,122 individuals aged 15–24 years from the Nepal Demographic and Health Survey (NDHS) 2022, focusing on a nationally representative sample. This study assessed the prevalence of sexual behaviors, including premarital sex, recent sexual activity, and multiple sexual partners. Determinants examined included socio-demographic characteristics, media use, smoking, and alcohol consumption. Bivariate and multivariate logistic regression analysis were conducted to determine the associations between social determinants and sexual behaviors.

### Results

The prevalence of premarital sex among the youth was 10.4%, with 15% of male youths reporting recent sexual activity and 2.8% having sexual intercourse with multiple partners. Variables significantly associated with premarital sex included older age groups (AOR = 2.81; 95% CI: 1.98–3.99), male (AOR = 7.87; 95% CI: 5.00–12.39), sales occupations (AOR = 2; 95% CI:1.12–3.57), smoking (AOR = 2.71; 95% CI:1.74–4.23), smokeless tobacco products (AOR = 1.94; 95% CI:1.12–3.34), and alcohol consumption (AOR = 2.97; 95% CI:2–4.41). Variables significantly associated with recent sexual activity included older age groups (AOR = 2.1; 95% CI:1.46, 3.03), being unmarried (AOR = 9.34; 95% CI:5.19–16.82), smoking (AOR = 2.01; 95% CI:1.33–3.05), use of smokeless products (AOR = 1.7; 95% CI:1.98–3.67), and alcohol consumption (AOR = 1.91; 95% CI:1.30–2.82). Youths using smokeless products had higher odds (AOR = 3.33; 95% CI:1.75–6.35) of having multiple sexual partners compared to those youths not using smokeless products.

**Data Availability Statement:** The data are available publicly in the open-access repository. The data can be downloaded from the official website of 'The

Demographic and Health Surveys' program. (https://dhsprogram.com/data/dataset/Nepal_Standard-DHS_2022.cfm?flag=0). The R script required to run the analysis is provided as a Supporting Information file.

**Funding:** The author(s) received no specific funding for this work.

**Competing interests:** The authors declared no potential conflicts of interest with respect to the research, authorship, and/or publication of this article

## Conclusion

Social determinants, along with smoking and alcohol consumption, were associated with sexual behaviors among youth. This study highlights the need for multicomponent health promotion (new public health) interventions which consider multi-level strategies, including culturally tailored sexual health programs, drug use behaviors, healthy lifestyle choices, comprehensive sexual health education for evidence-based interventions.

## Introduction

Certain sexual activities may increase the likelihood of sexually transmitted infections (STIs) and unintended pregnancies [1, 2] including having multiple sexual partners, participating in sexual activity under the influence of substances, and not using condoms [2, 3]. These behaviors can lead to reproductive health issues, such as STIs, human immunodeficiency virus (HIV) infections, unintended pregnancies, and abortions, while also contributing to psychological distress [1–3].

According to the United Nations, youth is defined as individuals aged between 15 and 24 years, representing a transitional phase from childhood dependence to adulthood independence [4]. Adolescents often engage in sexual activities at an early age, which become one of the major public health concerns among youth due to potential adverse outcomes [2, 5]. Various factors responsible to youths' engagement in sexual behavior, including biological factors such as age, sex; social factors such as peer pressure, substance abuse; and psychological factors such as low self-esteem and emotional distress [1–3, 6]. Limited knowledge about the consequences of sexual behavior, developmental stage, and limited access to health care services increase their vulnerability [2, 7]. The Nepal Demographic and Health Survey (NDHS) 2022 revealed that 46% of young women and 39% of young men aged 15–24 years have ever engaged in sexual intercourse. Among never married youths, 25% of men and 2% of women reported having had sex. Three percent of men had two or more sexual partners in the past twelve months, with 15% reporting sexual intercourse with individuals who were neither their wife nor in committed relationships [8].

Nepal has undergone significant demographic transformations, with a raising population of youths aged 15–24 years [4, 9]. As these youths transition into adulthood, they confronted challenges and opportunities related to their sexual well-being [10–12]. This demographic shift has raised a more open attitude towards sexual behaviors, leading to discussions about sexual health and, increasing their vulnerability to STIs/HIV [12]. A study conducted in Kathmandu, Nepal, found that 58% of adolescents had just a moderate knowledge of sexual and reproductive health. This status, however, provides opportunities for effective intervention [13, 14].

Factors such as gender norms and power dynamics greatly influences the sexual behavior of youths. These dynamics can result in situations where individuals, especially young women, have limited agency in making decisions about sex and contraception [14, 15]. Education and economic status are known as critical determinants of sexual behavior [3, 7, 16]. Access to quality education equips young individuals with the knowledge and skills necessary to make informed decisions about their sexual health [11, 15, 17]. Equally, economic vulnerability can impede access to healthcare services, influencing family planning and contraception decisions [7, 18].

Despite improvement in sexual education in Nepal, gaps persist in providing comprehensive information about safe sex practices, use of contraception, and sexual health in line with health promotion best practices [15, 19]. This lack of education contributes to misconceptions and risky behaviors among young people [19, 20]. In many parts of Nepal, especially remote or marginalized communities, access to quality healthcare services, including sexual and reproductive health services, remains limited. This lack of access may result in unprotected sex due to insufficient information, resources, and support [11, 19, 21]. Stigma surrounding discussions of sex can hinder open conversations, contributing in misinformation and risky behavior [21, 22]. With the increasing availability and accessibility of technology, including smartphones and social media, young people in Nepal are exposed to a wide range of information, some of which may be inaccurate or misleading. This digital landscape can significantly influence perceptions and behaviors regarding sex and relationships [23, 24]. Previous studies have documented that various factors influence sexual behavior among youth [1–3, 7], however, this study is significant as it examines the variations in sexual behaviors across different demographic groups, as well as the impact of risky behaviors such as smoking and alcohol use and media exposure [3, 5, 7, 10].

This study aims to explore the determinants of sexual behavior among Nepali youths, examining variations across demographic groups and the influence of risky behaviors such as smoking and alcohol use. The findings will have significant implications for policy and programmatic interventions aimed at promoting sexual health and reducing STI vulnerabilities.

## Materials and methods

### Data source

This study analyzed secondary data from the NDHS 2022, a nationally representative and internationally comparable descriptive cross-sectional household survey.

### Study settings

Nepal is a landlocked country located in South Asia, occupying an area of 147, 516 km$^2$. It consists of seven administrative provinces, which include 77 districts, six metropolitan cities, 11 sub-metropolitan cities, 276 urban municipalities, and 460 rural municipalities distributed across three ecological belts: mountain, hill, and terai.

### Sample size

The sample size and sampling technique of NDHS 2022 are described elsewhere [25]. The NDHS-2022 dataset included approximately 4,913 males and 14,845 females. For this study, data were extracted from NPMR82SV SPPS 20 version file, focusing on youth aged 15–24 years, resulting in a subset sample of 7,122 individuals (1,842 male youths and 5,280 female youths) were considered.

### Study variables

**Dependent variables.** The outcomes for this study were premarital sex, recent sexual activity, and multiple sexual partners. Premarital sex refers to never-married youths who have engaged in sexual intercourse. Recent sexual activity is considered as having sexual intercourse in the last 12 months with a partner who was not a wife or cohabitant. Multiple sexual partners is defined as having sexual intercourse with more than one partner in the past 12 months. Data on premarital sex encompasses a sample of 4426 male and female youths including those who

were unmarried, while recent sexual activity and multiple partners include the data of 1,842 male youths.

**Independent variables.** Independent variables included individual youth characteristics, household characteristics, and community characteristics. Individual youth characteristics include age (15–19 years and 20–24 years), sex (male and female) marital status (married and unmarried), education (no education, basic, secondary, higher), occupation (not working, agriculture, professional or technical or manager or clerical, sales and service, skilled or unskilled labor, others), used of media (yes, no), use of internet (yes, no), current smoking status (yes, no), use of smokeless products (yes, no) and alcohol consumption (yes, no). Household characteristics include religion (Buddhist, Christian, Hindu, Muslim, Others), ethnicity (Dalits, Janajati, Muslim, Other Terai caste, Upper Caste Groups), number of family members (1–2 members, 3–5 members, 6 members and above), and wealth quintile (poor, middle, and rich). The wealth index was modified from five wealth quintiles to three wealth quintiles for easy explanation, communication and data limitations. Community characteristics include area of residence (urban and rural), ecological region (mountain, hill and terai), province (Koshi, Madhesh, Bagmati, Gandaki, Lumbini, Karnali, and Sudurpaschhim)

## Statistical analysis

Data were extracted in SPSS version 20 and the SPSS datasets were imported to R statistical program for further analyses. The analyses included descriptive statistics, bivariate analysis, and logistic regression analysis. All data were weighted to account for the complex sample design, such as stratified sampling and unequal selection probabilities. The study utilized weighted techniques based on NDHS data to account for the complex survey design and ensure the representativeness of the analyses. The NDHS dataset provides sampling weights for each respondent, which was used to adjust for selection of probabilities at different stages of sampling, including stratification and clustering. These sampling weights are crucial for generating nationally representative estimates and accounting for oversampling in specific regions or population groups.

Pearson's Chi-squared test assessed children's characteristics, mother's characteristics, and household characteristics among children who were developmentally on track. Statistical analysis was conducted using R version 4.2.0 and RStudio 1314, employing the survey package for weighted analysis. Initially, each explanatory determinant were included in the model individually to examine its univariate relationship with the outcomes. Variable with significant univariate relationships (p < 0.05) were considered for inclusion in multivariate logistic regression models to identify key determinants.

The rationale behind, initially including variables one at a time helped avoid multicollinearity and assess each determinant's unique contribution to the outcome. This step-by-step approach identified potential confounders and interactions influencing the final multivariate models. Multivariate logistic regression models were then constructed to identify the most important determinants for each outcome, based on their theoretical relevance and statistical significance in univariate analysis. Adjusted odds ratios (AORs) and their 95% confidence intervals (95% CI) were calculated to depict the independent relationships between independent and dependent variables.

## Ethics statement

The primary data of the NDHS 2022 was approved by Nepal Health Research Council (NHRC) and made publicly available while ensuring participants anonymity. Secondary data was accessed through the DHS program.

## Results

Table 1 shows the socio-demographic characteristics of youths. The age distribution was evenly represented, with 51% in the 15–19 age group. Gender-wise, females were higher represented, comprising 74% of the population. Urban areas were slightly overrepresented, with 69% of youths residing in urban settings. Geographically, the majority of youths were from the terai region (57%). In terms of provinces, Madhesh, Bagmati, and Lumbini province had higher representation of youths. The religious composition was primarily Hindu with 82% of people falling in this group. Ethnicity, more than one third (34%) of youths belonged to the Janajati group. Educational attainment varied, with majority (60%) having completed secondary education. The occupational distribution illustrated a diverse range of employment statuses, with 40.7% reporting unemployment; the most common occupation reported was unskilled manual (6.8%). Family sizes were varied, with 38.1%) having six or more members. The wealth quintile showed a higher distribution among the rich (41.8%). In terms of health-related behaviors, the majority of youths did not engage in smoking (93%), use of smokeless products (94%), or consume alcohol (88%). Regarding sexual behavior, approximately 10.4% reported engaging in premarital sex, 15% reported recent sexual activity, and 2.8% reported having multiple sex partners.

Table 2 provides the results of the bivariate and multivariate analysis of factors associated with premarital sex among youths. The multivariate analysis identified several significant associations, including age, sex, ethnicity, occupation, number of family members, current smoking status, use of smokeless products, and alcohol consumption. Youths from the age group of 20–24 years had higher odds (AOR = 2.81; 95% CI: 1.98–3.99) of engaging in premarital sex compared to those youths from the 15–19 years age group. Male youths were 7.87 times more likely to have premarital sex than female youths (AOR = 7.87; 95% CI: 5.00–12.39). Those from other terai caste (Teli, Kalwar, Kurmi and others) had lower odds to have premarital sex (AOR = 0.3; 95% CI:0.16–0.56) compared to youth from Dalit ethnicity. Youths employed in sales were twice as likely to engage in premarital sex compared to those youths in agriculture (AOR = 2; 95% CI:1.12–3.57). Moreover, youths who belonged to the family with more family members were less likely to have premarital sex; those from families with 3–5 family members had lower odds (AOR = 0.52; 95% CI:0.31–0.85), and those with six or more members also had lower odds (AOR = 0.5; 95% CI:0.3–0.84) compared to youths from families with one to two family members. Youth who were current smokers had higher odds (AOR = 2.71; 95% CI:1.74–4.23) of having premarital sex compared to who did not smoke. Youth who were using smokeless products were 1.94 times more likely to have premarital sex (AOR = 1.94; 95% CI:1.12–3.34) than those youth who were not using such products. Youth who consumed alcohol were 2.97 times more likely to engage premarital sex (AOR = 2.97; 95% CI:2–4.41) than those who did not consume alcohol.

Table 3 shows the results of the bivariate and multivariate analysis of factors associated with recent sexual activity among male youths. The multivariate analysis identified significant associations with age, ethnicity, marital status, current smoking status, use of smokeless products, and alcohol consumption. Youths from the age group of 20–24 years were 2.11 times more likely to engage in recent sexual activity than to those youths from the age group of 15–19 years (AOR = 2.11; 95% CI:1.46–3.03). Those belonging to other terai caste were 30% less likely to engage in recent sexual activity (AOR = 0.3; 95% CI:0.15–0.56) compared to youths from Dalit ethnicity. The male youths who were unmarried were 9.34 times more likely to have engaged in recent sexual activity (AOR = 9.34; 95% CI:5.19–16.82) than the male youths who were married. Youth who were current smokers were 2 times more likely to have recent sexual activity (AOR = 2; 95% CI: 1.33, 3.05) than those youth who were non-smokers. Youths using

**Table 1. Socio demographic characteristics of youths (N = 7,122).**

| Characteristic | Overall | Percentage |
|---|---|---|
| | **N = 7122** | **(%)** |
| **Age** | | |
| 15–19 years | 3,627 | 51 |
| 20–24 years | 3,495 | 49 |
| **Sex** | | |
| Male | 1,842 | 26 |
| Female | 5,280 | 74 |
| **Residence** | | |
| Rural | 2,205 | 31 |
| Urban | 4,917 | 69 |
| **Ecological Region** | | |
| Mountain | 355 | 5 |
| Hill | 2,740 | 38 |
| Terai | 4,028 | 57 |
| **Province** | | |
| Koshi | 1,166 | 16 |
| Madhesh | 1,648 | 23 |
| Bagmati | 1,423 | 20 |
| Gandaki | 586 | 8.2 |
| Lumbini | 1,192 | 17 |
| Karnali | 478 | 6.7 |
| Sudurpashchim | 630 | 8.8 |
| **Religion** | | |
| Buddhist | 473 | 6.6 |
| Christian | 215 | 3 |
| Hindu | 5,862 | 82 |
| Muslim | 399 | 5.6 |
| Others | 174 | 2.4 |
| **Ethnicity** | | |
| Dalits | 1,149 | 16 |
| Janajati | 2,439 | 34 |
| Muslim | 394 | 5.5 |
| Other Terai caste | 1,200 | 17 |
| Upper Caste Groups | 1,941 | 27 |
| **Education** | | |
| No education | 410 | 5.8 |
| Basic | 2,287 | 32 |
| Secondary | 4,271 | 60 |
| Higher | 154 | 2.2 |
| **Occupation** | | |
| Agriculture | 373 | 5.2 |
| Clerical | 116 | 1.6 |
| Not working | 2,901 | 40.7 |
| Professional/technical/managerial | 310 | 4.3 |
| Sales | 451 | 6.3 |
| Skilled manual | 469 | 6.5 |
| Unskilled manual | 484 | 6.8 |

*(Continued)*

**Table 1.** (Continued)

| Characteristic | Overall | Percentage |
|---|---|---|
| | **N = 7122** | **(%)** |
| Unknown | 2018 | 28.3 |
| **Family members** | | |
| 1–2 members | 525 | 7.3 |
| 3–5 members | 2,458 | 34.5 |
| 6 or more members | 2,715 | 38.1 |
| Unknown | 1423 | 19.9 |
| **Wealth Quintile** | | |
| Poor | 2,730 | 38.3 |
| Middle | 1,414 | 19.8 |
| Rich | 2,978 | 41.8 |
| **Use of Media** | | |
| No | 1,336 | 19 |
| Yes | 5,786 | 81 |
| **Use of internet** | | |
| No | 1,305 | 18 |
| Yes | 5,817 | 82 |
| **Current Smoking** | | |
| No | 6,612 | 93 |
| Yes | 510 | 7.2 |
| **Use of Smokeless products** | | |
| No | 6,691 | 94 |
| Yes | 431 | 6 |
| **Alcohol Consumption** | | |
| No | 6,270 | 88 |
| Yes | 712 | 10 |
| Unknown | 140 | 1.9 |
| **Premarital Sex (N = 4426)** | | |
| Yes | 462 | 10.4 |
| No | 3964 | 89.6 |
| **Multiple Sex Partners (N = 1,842)** | | |
| Yes | 51 | 2.8 |
| No | 1791 | 97.2 |
| **Recent Sexual Activity (N = 1,842)** | | |
| Yes | 278 | 15 |
| No | 1564 | 85 |

smokeless products were more likely to have recent sexual activity (AOR = 1.7; 95% CI: 1.33, 3.05) compared to those youth who did not used smokeless products. Youth, those who consumed alcohol were almost twice more likely to have recent sexual activity (AOR = 1.9; 95% CI: 1.30, 2.82) compared to those youth who did not consume alcohol.

Table 4 presents the results of the bivariate and multivariate analysis of factors associated with having multiple sexual partners among male youths. In the multivariate analysis, ethnicity, and the use of smokeless products were significantly associated with having multiple sexual partners. The youths belonging to other terai caste had lower odds (AOR = 0.022; 95% CI:0.03–0.77) of having multiple sex partner compared to those from the Dalit ethnicity.

**Table 2. Factors associated with premarital sex among youths (N = 4426).**

| Characteristic | Yes | Total | Unadjusted | | | Adjusted | | |
|---|---|---|---|---|---|---|---|---|
| | (%) | (n = 462) | OR | 95% CI | p- value | OR | 95% CI | p- value |
| **Age** | | | | | | | | |
| 15–19 years | 5.6 | 168 | 1 | | | 1 | | |
| 20–24 years | 20.9 | 294 | 4.49 | 3.56, 5.66 | <**0.001** | 2.8 1 | 1.98, 3.99 | <**0.001** |
| **Sex** | | | | | | | | |
| Female | 3.0 | 88 | 1 | | | 1 | | |
| Male | 25.0 | 374 | 10.78 | 8.19, 14.1 | <**0.001** | 7.8 7 | 5.00, 12.3 | <**0.001** |
| **Residence** | | | | | | | | |
| Rural | 10.3 | 130 | 1 | | | | | |
| Urban | 10.5 | 332 | 1.02 | 0.82, 1.26 | 0.89 | | | |
| **Ecological Region** | | | | | | | | |
| Mountain | 10.6 | 20 | 1 | | | | | |
| Hill | 11.6 | 214 | 1.11 | 0.69, 1.77 | 0.66 | | | |
| Terai | 9.5 | 228 | 0.89 | 0.56, 1.42 | 0.63 | | | |
| **Province** | | | | | | | | |
| Koshi | 9.4 | 68 | 1 | | | | | |
| Madhesh | 7.1 | 60 | 0.74 | 0.49, 1.11 | 0.14 | | | |
| Bagmati | 12.9 | 141 | 1.43 | 0.98, 2.07 | 0.06 | | | |
| Gandaki | 12.7 | 50 | 1.4 | 0.93, 2.12 | 0.11 | | | |
| Lumbini | 10.0 | 74 | 1.07 | 0.73, 1.57 | 0.72 | | | |
| Karnali | 12.7 | 32 | 1.4 | 0.95, 2.06 | 0.09 | | | |
| Sudurpashchim | 9.6 | 37 | 1.02 | 0.69, 1.52 | 0.91 | | | |
| **Religion** | | | | | | | | |
| Buddhist | 15.2 | 50 | 1 | | | 1 | | |
| Christian | 12.5 | 16 | 0.8 | 0.39, 1.64 | 0.54 | 1.43 | 0.50, 4.12 | 0.51 |
| Hindu | 10.1 | 366 | 0.62 | 0.41, 0.95 | **0.03** | 1.02 | 0.55, 1.89 | 0.95 |
| Muslim | 8.7 | 20 | 0.53 | 0.27, 1.07 | 0.08 | 0.93 | 0.34, 2.58 | 0.9 |
| Others | 9.0 | 10 | 0.55 | 0.25, 1.24 | 0.15 | 0.46 | 0.13, 1.61 | 0.23 |
| **Ethnicity** | | | | | | | | |
| Dalits | 11.5 | 68 | 1 | | | 1 | | |
| Janajati | 13.2 | 210 | 1.18 | 0.84, 1.66 | 0.35 | 0.9 5 | 0.55, 1.67 | 0.87 |

(*Continued*)

**Table 2.** (Continued)

| Characteristic | Yes | Total | Unadjusted | | | Adjusted | | |
|---|---|---|---|---|---|---|---|---|
| | (%) | (n = 462) | OR | 95% CI | p- value | OR | 95% CI | p- value |
| Muslim | 8.9 | 20 | 0.75 | 0.40, 1.43 | 0.38 | | | |
| Other Terai caste | 5.9 | 38 | 0.48 | 0.30, 0.77 | **<0.001** | 0.3 | 0.16, 0.56 | **<0.001** |
| Upper Caste Groups | 9.2 | 126 | 0.78 | 0.55, 1.13 | 0.19 | 0.9 | 0.51, 1.61 | 0.73 |
| **Education** | | | | | | | | |
| No education | 5.3 | 6 | 1 | | | 1 | | |
| Basic | 10.2 | 119 | 2.01 | 0.70, 5.78 | 0.19 | 1.01 | 0.23, 4.45 | 0.99 |
| Secondary | 10.4 | 317 | 2.07 | 0.73, 5.88 | 0.17 | 1.32 | 0.29, 5.92 | 0.72 |
| Higher | 17.4 | 20 | 3.75 | 1.14, 12.29 | **0.03** | 1.44 | 0.26, 7.86 | 0.67 |
| **Occupation (n = 434)** | | | | | | | | |
| Agriculture | 19.4 | 59 | 1 | | | 1 | | |
| Clerical | 7.2 | 6 | 0.32 | 0.11, 0.96 | **0.04** | 0.69 | 0.19, 2.42 | 0.56 |
| Not working | 5.5 | 108 | 0.24 | 0.17, 0.35 | **<0.001** | 0.72 | 0.45, 1.17 | 0.19 |
| Professional/technical/managerial | 15.6 | 36 | 0.77 | 0.46, 1.30 | 0.33 | 1.38 | 0.70, 2.72 | 0.36 |
| Sales | 20.4 | 63 | 1.76 | 1.14, 2.71 | **0.01** | 2 | 1.12, 3.57 | **0.02** |
| Skilled manual | 29.7 | 84 | 1.07 | 0.68, 1.69 | 0.77 | 1.54 | 0.82, 2.88 | 0.18 |
| Unskilled manual | 23.9 | 78 | 1.31 | 0.88, 1.96 | 0.19 | 1.16 | 0.66, 2.03 | 0.61 |
| **Family members (n = 387)** | | | | | | | | |
| 1–2 members | 19.7 | 64 | 1 | | | 1 | | |
| 3–5 members | 11.2 | 186 | 0.51 | 0.35, 0.76 | **<0.001** | 0.52 | 0.31, 0.85 | **0.01** |
| 6 or more members | 9.1 | 137 | 0.41 | 0.27, 0.61 | **<0.001** | 0.5 | 0.30, 0.84 | **0.009** |
| **Wealth Quintile** | | | | | | | | |
| Middle | 9.8 | 82 | 1 | | | | | |
| Poor | 10.1 | 148 | 1.04 | 0.77, 1.39 | 0.81 | | | |
| Rich | 11.0 | 232 | 1.14 | 0.84, 1.54 | 0.4 | | | |
| **Use of Media** | | | | | | | | |
| No | 12.3 | 86 | 1 | | | | | |
| Yes | 10.1 | 376 | 0.8 | 0.59, 1.09 | 0.16 | | | |
| **Use of Internet** | | | | | | | | |
| No | 4.8 | 33 | 1 | | | 1 | | |

(*Continued*)

**Table 2.** (Continued)

| Characteristic | Yes | Total | Unadjusted | | | Adjusted | | |
|---|---|---|---|---|---|---|---|---|
| | (%) | (n = 462) | OR | 95% CI | p- value | OR | 95% CI | p- value |
| Yes | 11.5 | 429 | 2.58 | 1.77, 3.76 | <0.001 | 0.93 | 0.51, 1.71 | 0.82 |
| **Current Smoking** | | | | | | | | |
| No | 7.2 | 294 | 1 | | | 1 | | |
| Yes | 48.9 | 168 | 12.3 | 9.18, 16.49 | <0.001 | 2.71 | 1.74, 4.23 | <0.001 |
| **Use of Smokeless products** | | | | | | | | |
| No | 8.5 | 353 | 1 | | | 1 | | |
| Yes | 43.6 | 109 | 8.38 | 6.03, 11.64 | <0.001 | 1.94 | 1.12, 3.34 | 0.017 |
| **Alcohol Consumption** | | | | | | | | |
| No | 6.5 | 253 | 1 | | | 1 | | |
| Yes | 39.6 | 196 | 9.4 | 7.25, 12.18 | <0.001 | 2.97 | 2.00, 4.41 | <0.001 |

Youths using smokeless products were more likely to have multiple sex partner (AOR = 3.33; 95% CI:1.75–6.35) compared to those youths who did not use such products.

## Discussion

The study findings identified a significant proportion of youths engaging in premarital sex, with a notable prevalence among male who reported recent sexual activity and multiple sexual partners. It was found that 2.8% of the female youths and 25% male youths reported engaging in premarital sex. There was a significant increase in premarital sexual activity among females from 2016, when the prevalence was 0.6%, while the prevalence among males remained relatively stagnant at 25.4% in 2016 [8]. The rise in premarital sexual activity among females can be attributed to changing societal norms, improved education and awareness, urbanization, economic independence, delayed marriages, peer influence, access to contraceptives, and women's empowerment in recent years [12, 14, 15, 18]. Moreover, the study indicates that 15% of male youths have been involved in recent sexual activity, that was similar to findings from NHDS 2016 data (15.8%).These similarities can be attributed to cultural and social norms, a lack of comprehensive sexual education, and limited access to youth-friendly sexual and reproductive health services, all of which contribute to sustained patterns of sexual behavior among male youth over time. However, the proportion of male youths reporting multiple sexual partners was 2.8%, a decrease from 4% based on NHDS 2016 data. This decline suggest that comprehensive sex education has effectively conveyed the importance of responsible sexual behavior to male youths, emphasizing the benefits of monogamous relationships and raising awareness about STI/HIV and the risks associated with multiple partners [19, 21]. Male youths may be placing greater emphasis on the quality and stability of relationships rather than the quantity of partners. This shift potentially influenced by changing societal attitudes towards commitment and long-term partnerships [14, 20, 26]. Our findings suggest that while females are exploring premarital relationships, and male are becoming more cautious concerning engagement in multiple sexual partnerships. These distinguishing findings highlight the need for comprehensive awareness programs that addresses the evolving dynamics of sexual relationships for both sex. Despite existing laws intended to protect minors from engaging in

**Table 3. Factors associated with recent sex among male youth (N = 1,842).**

| Characteristic | Yes (%) | Total (n = 278) | Unadjusted | | | Adjusted | | |
|---|---|---|---|---|---|---|---|---|
| | | | OR | 95% CI | p-value | OR | 95% CI | p-value |
| **Age** | | | | | | | | |
| 15–19 years | 10.1 | 99 | 1 | | | 1 | | |
| 20–24 years | 20.9 | 179 | 4.5 | 3.56, 5.66 | <**0.001** | 2.1 | 1.46, 3.03 | <**0.001** |
| **Residence** | | | | | | | | |
| Rural | 15.6 | 81 | 1 | | | | | |
| Urban | 3.0 | 197 | 1 | 0.82, 1.26 | 0.89 | | | |
| **Ecological Region** | | | | | | | | |
| Mountain | 18.4 | 15 | 1 | | | | | |
| Hill | 17.4 | 127 | 1.1 | 0.69, 1.77 | 0.66 | | | |
| Terai | 13.2 | 136 | 0.9 | 0.56, 1.42 | 0.63 | | | |
| **Province** | | | | | | | | |
| Koshi | 12.2 | 38 | 1 | | | | | |
| Madhesh | 7.8 | 34 | 0.7 | 0.49, 1.11 | 0.14 | | | |
| Bagmati | 19.1 | 87 | 1.4 | 0.98, 2.07 | 0.06 | | | |
| Gandaki | 19.3 | 24 | 1.4 | 0.93, 2.12 | 0.11 | | | |
| Lumbini | 21 | 58 | 1.1 | 0.73, 1.57 | 0.72 | | | |
| Karnali | 16.7 | 17 | 1.4 | 0.95, 2.06 | 0.09 | | | |
| Sudurpashchim | 14.4 | 20 | 1 | 0.69, 1.52 | 0.91 | | | |
| **Religion** | | | | | | | | |
| Buddhist | 18.2 | 27 | 1 | | | 1 | | |
| Christian | 9 | 5 | 0.8 | 0.39, 1.64 | 0.54 | 0.7 | 0.15, 3.16 | 0.63 |
| Hindu | 15.3 | 227 | 0.6 | 0.41, 0.95 | **0.03** | 1.3 | 0.60, 2.63 | 0.55 |
| Muslim | 14.2 | 16 | 0.5 | 0.27, 1.07 | 0.08 | 1.3 | 0.43, 3.90 | 0.65 |
| Others | 8.3 | 4 | 0.6 | 0.25, 1.24 | 0.15 | 0.4 | 0.13, 1.39 | 0.16 |
| **Ethnicity** | | | | | | | | |
| Dalits | 16.2 | 44 | 1 | | | 1 | | |
| Janajati | 18.5 | 119 | 1.2 | 0.84, 1.66 | 0.35 | 1 | 0.59, 1.73 | 0.98 |
| Muslim | 14.2 | 16 | 0.8 | 0.40, 1.43 | 0.38 | 0.9 | 0.56, 2.43 | 0.48 |
| Other Terai caste | 6.6 | 23 | 0.5 | 0.30, 0.77 | < **0.001** | 0.3 | 0.15, 0.56 | <**0.001** |

*(Continued)*

**Table 3.** (Continued)

| Characteristic | Yes (%) | Total (n = 278) | Unadjusted | | | Adjusted | | |
|---|---|---|---|---|---|---|---|---|
| | | | OR | 95% CI | p-value | OR | 95% CI | p-value |
| Upper Caste Groups | 16.2 | 78 | 0.8 | 0.55, 1.13 | 0.19 | 0.8 | 0.47, 1.42 | 0.47 |
| **Marital Status** | | | | | | | | |
| Married | 5.6 | 19 | 1 | | | 1 | | |
| Unmarried | 17.3 | 259 | 3.6 | 2.13, 5.92 | **<0.001** | 9.3 | 5.19, 16.82 | **<0.001** |
| **Education** | | | | | | | | |
| No education | 11.9 | 6 | 1 | | | 1 | | |
| Basic | 10.6 | 65 | 1.4 | 0.47, 4.24 | 0.54 | 0.7 | 0.20, 2.61 | 0.61 |
| Secondary | 17 | 193 | 2.1 | 0.72, 6.29 | 0.17 | 1.6 | 0.43, 5.61 | 0.5 |
| Higher | 31.9 | 14 | 4.3 | 1.16, 16.16 | **0.03** | 2.4 | 0.52, 10.79 | 0.26 |
| **Occupation** | | | | | | | | |
| Agriculture | 11.9 | 45 | 1 | | | 1 | | |
| Clerical | 32 | 6 | 3.5 | 1.03, 11.74 | **0.04** | 2.2 | 0.63, 7.83 | 0.21 |
| Not working | 9.2 | 55 | 0.8 | 0.47, 1.19 | 0.22 | 0.6 | 0.39, 1.04 | 0.071 |
| Professional/technical/man agerial | 23.6 | 21 | 2.3 | 1.12, 4.65 | **0.02** | 1.3 | 0.60, 2.66 | 0.53 |
| Sales | 21.7 | 35 | 2 | 1.17, 3.57 | **0.01** | 1.3 | 0.71, 2.39 | 0.39 |
| Skilled manual | 21.6 | 63 | 2 | 1.26, 3.29 | **<0.001** | l.7 | 0.95, 3.11 | 0.073 |
| Unskilled manual | 17.1 | 54 | 1.5 | 0.97, 2.39 | 0.07 | 1.4 | 0.80, 2.28 | 0.27 |
| **Family Members** | | | | | | | | |
| 1–2 members | 21.1 | 33 | 1 | | | | | |
| 3–5 members | 15 | 109 | 0.7 | 0.38, 1.16 | 0.15 | | | |
| 6 or more members | 14.2 | 86 | 0.6 | 0.35, 1.10 | 0.1 | | | |
| **Wealth Quintile** | | | | | | | | |
| Middle | 12.2 | 43 | 1 | | | | | |
| Poor | 14 | 90 | 1.2 | 0.78, 1.76 | 0.44 | | | |
| Rich | 17.2 | 145 | 1.5 | 0.99, 2.27 | 0.06 | | | |
| **Use of Media** | | | | | | | | |
| No | 14 | 46 | 1 | | | | | |
| Yes | 15.3 | 233 | 1.1 | 0.70, 1.76 | 0.65 | | | |
| **Use of Internet** | | | | | | | | |
| No | 11 | 17 | l | | | | | |

(*Continued*)

**Table 3.** (Continued)

| Characteristic | Yes (%) | Total (n = 278) | Unadjusted | | | Adjusted | | |
|---|---|---|---|---|---|---|---|---|
| | | | OR | 95% CI | p-value | OR | 95% CI | p-value |
| Yes | 15.5 | 261 | 1.5 | 0.84, 2.64 | 0.17 | | | |
| **Current Smoking** | | | | | | | | |
| No | 1 l.5 | 160 | l | | | I | | |
| Yes | 26.4 | 118 | 2.8 | 2.01, 3.80 | <**0.001** | 2 | 1.33, 3.05 | **0.001** |
| **Use of Smokeless products** | | | | | | | | |
| No | 13.2 | 198 | 1 | | | 1 | | |
| Yes | 23.3 | 80 | 2 | 1.41, 2.82 | <**0.001** | 1.7 | 1.04, 2.79 | **0.035** |
| **Alcohol Consumption** | | | | | | | | |
| No | 11.I | 146 | I | | | 1 | | |
| Yes | 25.1 | 132 | 2.7 | 1.98, 3.67 | <**0.001** | 1.9 | 1.30, 2.82 | **0.001** |

sexual activities [27, 28], the persistently high prevalence of sexual activities necessities targeted educational programs aimed at adolescents and youths population.

In the multivariate analysis, age was associated with premarital sex and recent sexual activity. Regarding age differences (15–19 versus 20–24 years), this study confirmed that youths, who were involved in premarital sexual intercourse and recent sexual activity are comparatively more prevalent among the older age group (20–24 years). This finding aligns with similar studies [3, 5–7, 16, 17], emphasizing that older youth may face increased exposure to factors influencing sexual behavior. Social norms and peer influences also play significant roles, with older adolescents often feeling more societal pressure to engage in sexual relationships. Moreover, they are characterized by a greater emotional and physical intimacy and a preference for stable relationships or cohabitation, compared to younger adolescents who may be exploring romantic relationships for the first time [5, 10, 12].

This study has shown that male youths were more likely to be involved in premarital sex than female youths. This result is consistent with previous studies indicating that young males were more likely to have premarital sex than young females [1, 4, 16, 29]. The reason behind this might be that Nepal is a patriarchal country where males generally enjoy greater freedom compared to females [9, 10, 14]. Males can enjoy their gender privilege, while females are often expected to adhere to social norms and beliefs which oppose premarital sexual activity. Females are expected to maintain their virginity till marriage, as their virginity is greatly valued [9, 15, 30]. In addition, while considering the consequences of sexual activity, females are at higher risk of experiencing sexual health consequences like acquiring STI/HIV, teenage pregnancy, and unsafe abortion, which may make them less likely to engage in premarital sex due to the potential for unnecessary troubles and various complications [17, 30, 31]. However, underreporting of the private sexual activity by females might also be another determinant to this disparity [32].

Ethnicity was found to be associated with all sexual risk behaviors (premarital sex, recent sexual activity, and multiple partners), as the youths belonging to other terai castes were found to be less likely to be engaged in sexual risk behavior. Traditional cultural norms and values within these ethnic groups emphasize stricter social norms and expectations regarding sexual

**Table 4. Factors associated with multiple sexual partners among male youth (N = 1,842).**

| Characteristic | Yes (%) | Total (n = 51) | Unadjusted | | | Adjusted | | |
|---|---|---|---|---|---|---|---|---|
| | | | OR | 95% CI | p-value | OR | 95% CI | p-value |
| **Age** | | | | | | | | |
| 15–19 years | 1.6 | 16 | 1 | | | 1 | | |
| 20–24 years | 4.1 | 35 | 2.54 | 1.23, 5.23 | **0.01** | 1.57 | 0.79, 3.14 | 0.2 |
| **Residence** | | | | | | | | |
| Rural | 2.2 | 11 | 1 | | | | | |
| Urban | 3.0 | 40 | 1.4 | 0.72, 2.70 | 0.32 | | | |
| **Ecological Region** | | | | | | | | |
| Mountain | 3.8 | 3 | 1 | | | | | |
| **Hill** | 2.9 | 21 | 0.77 | 0.29, 2.05 | 0.61 | | | |
| Terai | 2.6 | 27 | 0.68 | 0.28, 1.66 | 0.39 | | | |
| **Province** | | | | | | | | |
| Koshi | 2.4 | 7 | 1 | | | | | |
| Madhesh | 1.5 | 7 | 0.64 | 0.19, 2.10 | 0.46 | | | |
| Bagmati | 3.6 | 16 | 1.54 | 0.51, 4.61 | 0.44 | | | |
| Gandaki | 1.8 | 2 | 0.76 | 0.18, 3.25 | 0.71 | | | |
| Lumbini | 4.1 | 11 | 1.79 | 0.65, 4.93 | 0.26 | | | |
| Karnali | 2.9 | 3 | 1.26 | 0.42, 3.77 | 0.68 | | | |
| Sudurpashchim | 3.0 | 4 | 1.29 | 0.43, 3.91 | 0.65 | | | |
| **Religion** | | | | | | | | |
| Buddhist | 2.3 | 3 | | | | | | |
| Christian | 0 | 0 | | | | | | |
| Hindu | 3.0 | 45 | | | | | | |
| Muslim | 2.8 | 3 | | | | | | |
| Others | 0 | 0 | | | | | | |
| **Ethnicity** | | | | | | | | |
| Dalits | 4.6 | 12 | 1 | | | 1 | | |
| Janajati | 3.2 | 20 | 0.68 | 0.29, 1.58 | 0.37 | 0.73 | 0.31, 1.71 | 0.47 |
| Muslim | 2.8 | 3 | 0.58 | 0.12, 2.79 | 0.5 | 0.82 | 0.18, 3.70 | 0.79 |
| Other Terai caste | 0.7 | 2 | 0.14 | 0.03, 0.68 | **0.01** | 0.16 | 0.03, 0.77 | **0.022** |
| Upper Caste Groups | 2.7 | 13 | 0.56 | 0.21, 1.51 | 0.25 | 0.86 | 0.31, 2.44 | 0.78 |
| **Marital Status** | | | | | | | | |
| Married | 4.3 | 15 | | | | | | |

*(Continued)*

**Table 4.** (Continued)

| Characteristic | Yes (%) | Total (n = 51) | Unadjusted | | | Adjusted | | |
|---|---|---|---|---|---|---|---|---|
| | | | OR | 95% CI | p-value | OR | 95% CI | p-value |
| Unmarried | 2.4 | 36 | | | | | | |
| **Education** | | | | | | | | |
| No education | 0.0 | 0 | | | | | | |
| Basic | 2.8 | 17 | | | | | | |
| Secondary | 2.6 | 30 | | | | | | |
| Higher | 9.6 | 4 | | | | | | |
| **Occupation** | | | | | | | | |
| Agriculture | 1.3 | 5 | | | | | | |
| Clerical | 0.0 | 0 | | | | | | |
| Not working | 2.3 | 14 | | | | | | |
| Professional/technical/man agerial | 4.4 | 4 | | | | | | |
| Sales | 3.4 | 6 | | | | | | |
| Skilled manual | 3.8 | 11 | | | | | | |
| Unskilled manual | 3.9 | 12 | | | | | | |
| **Family Members** | | | | | | | | |
| 1–2 members | 2.2 | 3 | 1 | | | | | |
| 3–5 members | 3.2 | 23 | 1.45 | 0.25, 8.39 | 0.68 | | | |
| 6 or more members | 2.7 | 16 | 1.23 | 0.21, 7.24 | 0.82 | | | |
| **Children** | | | | | | | | |
| More than one child | 5.5 | 3 | 1 | | | | | |
| No children | 2.7 | 45 | 0.47 | 0.14, 1.58 | 0.22 | | | |
| One child | 2.8 | 4 | 0.49 | 0.11, 2.17 | 0.34 | | | |
| **Wealth Quintile** | | | | | | | | |
| Middle | 2.5 | 9 | 1 | | | | | |
| Poor | 2.3 | 15 | 0.91 | 0.40, 2.08 | 0.83 | | | |
| Rich | 3.3 | 28 | 1.32 | 0.56, 3.12 | 0.52 | | | |
| **Use of Media** | | | | | | | | |
| No | 3.0 | 10 | 1 | | | | | |
| Yes | 2.7 | 41 | 0.89 | 0.34, 2.37 | 0.82 | | | |
| **Use of Internet** | | | | | | | | |
| No | 1.0 | 2 | 1 | | | | | |
| Yes | 2.9 | 50 | 3.1 | 0.69, 13.80 | 0.14 | | | |
| **Current Smoking** | | | | | | | | |
| No | 1.8 | 25 | 1 | | | 1 | | |
| Yes | 5.9 | 26 | 3.52 | 1.77, 6.98 | **<0.001** | 1.65 | 0.74, 3.70 | 0.22 |
| **Use of Smokeless products** | | | | | | | | |
| No | 1.6 | 24 | 1 | | | 1 | | |

(*Continued*)

**Table 4.** (Continued)

| Characteristic | Yes (%) | Total (n = 51) | Unadjusted | | | Adjusted | | |
|---|---|---|---|---|---|---|---|---|
| | | | OR | 95% CI | p-value | OR | 95% CI | p-value |
| Yes | 8.0 | 27 | 5.44 | 2.76, 10.74 | <0.001 | 3.33 | 1.75, 6.35 | <0.001 |
| **Alcohol Consumption** | | | | | | | | |
| No | 1.6 | 21 | 1 | | | 1 | | |
| Yes | 5.7 | 30 | 3.7 | 1.87, 7.35 | <0.001 | 1.96 | 0.91, 4.21 | 0.083 |

conduct. Additionally, the influence of cultural practice and community norms may play a role in discouraging premarital sex and sexual activity with multiple partners [14, 32, 33].

Youths employed in sales were more likely to be involved in premarital sex compared to those working in agriculture. Studies found that youths engaged in sales are exposed to large numbers of customers and people from different places and backgrounds [15, 20, 34] and their roles require frequent interaction with customers, people who are extroverted and possess strong communication skills are typically hired [34, 35]. The nature of their job and interactions with people may facilitate the formation of sexual partners and promote sexual activity and have active in sex life [10, 12, 34, 35]. Targeted interventions, such as awareness programs, could be provided to such youths, utilizing national and regional census data that include information about their occupation.

The findings of this study have shown that an increase in family size decreases the likelihood of engaging premarital sex. A possible explanation for this finding is that larger family size tends to result in increased supervision of individual family members, placing youths under the control of their elder household members, like parents, grand-parents, or guardians, which will hinder them from engaging in any risk taking behavior, including sexual activity [3, 26, 36]. Moreover, having many family members may allow them to spend more time with their families, and so reducing their time with peers and lowering the influence of peer pressure [26, 36]. Similar findings have been shown in studies indicating that youths living without their family or independently from their families are more likely to have sexual behavior compared to those who live with their family members [29, 36], as they enjoy greater freedom from the misunderstanding of older household members, leading to increase participation in various risky behaviors including early sexual activity.

Consistent with many studies [3, 5, 6, 16, 35], this study demonstrates a relationship between the current smoking status of youths and premarital sex as well as recent sexual activity. Additionally, the use of smokeless products was also found to be associated with premarital sex, recent sexual activity, and multiple sexual partners. Therefore, the use of any tobacco products, either cigarettes or electronic cigarettes (vapes), or chewing tobacco, can impair judgment, leading to riskier decision-making and engagement in sexual behavior [6, 7, 17]. Furthermore, social and cultural factors, such as peer influence and norms within tobacco-using communities, could contribute to higher rates of sexual activity among users of these products [14, 15, 18].

This study revealed that youth consuming alcohol increases the likelihood of involving in sexual behavior, which is similar to the findings of previous studies [3, 33, 37]. The possible explanation may be that consumption of alcohol impairs mental capabilities, inhibiting the ability to recognize risks related to sexual behavior [33, 38]. Alcohol consumption is often intertwined with social settings and peer interactions, where norms surrounding sexual

behavior may be less restrained. Moreover, alcohol consumption decreases self-control, which may lead to increased sexual activity among male youths. Likewise, young people generally use alcohol as a strategy to facilitate sexual meets [33, 37, 38].

This study had few limitations. The cross-sectional nature of the data collection limits the power to establish a causal relationship. This study also did not include potentially significant variables such as parental influence, substance use, survival sex due to lack of data limitations, which could provide a comprehensive understanding of the factors influencing sexual behavior among youths. Despite these limitations, the study possesses certain strengths, including, the use of a nationally representative dataset, enhancing the generalizability of the findings to the broader youth population in Nepal.

## Conclusion

This study determines vital insights into how socio-demographic variables and risk behaviors are associated with sexual behaviors among youths, presenting the need for targeted interventions. These findings highlighted the importance of behavior change communication using socioecological and health promotion models. This study recommends increasing health literacy, creating an enabling environment for safer sexual behaviors, and playing the mediating roles of different public, private, health, and non-health stakeholders.

## Supporting information

**S1 File.**
(RCM)

## Acknowledgments

Authors would like to acknowledge the DHS program for allowing us to use the NDHS 2022 data.

## Author Contributions

**Conceptualization:** Grishu Shrestha, Sampurna Kakchapati, Sushil Chandra Baral.

**Data curation:** Reecha Piya.

**Formal analysis:** Reecha Piya, Sampurna Kakchapati, Parash Mani Sapkota.

**Investigation:** Deepak Joshi.

**Methodology:** Sampurna Kakchapati, Parash Mani Sapkota.

**Project administration:** Grishu Shrestha.

**Resources:** Deepak Joshi.

**Supervision:** Deepak Joshi, Sushil Chandra Baral.

**Validation:** Deepak Joshi, Sushil Chandra Baral.

**Visualization:** Parash Mani Sapkota.

**Writing – original draft:** Grishu Shrestha, Reecha Piya, Deepak Joshi.

**Writing – review & editing:** Grishu Shrestha, Sampurna Kakchapati, Sushil Chandra Baral.

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
