## [Decision Letter · Decision Letter 0]

20 Feb 2024

PONE-D-24-00348Exploring the Drivers of Risky Sexual Behavior and its social determinants in Nepali Youth: Insights from NDHS 2022PLOS ONE

Dear Dr. Kakchapati,

Thank you for submitting your manuscript to PLOS ONE. After careful consideration, we feel that it has merit but does not fully meet PLOS ONE’s publication criteria as it currently stands. Therefore, we invite you to submit a revised version of the manuscript that addresses the points raised during the review process.

We look forward to receiving your revised manuscript.

Kind regards,

Shalik Ram Dhital, PhD

Academic Editor

PLOS ONE

2. PLOS requires an ORCID iD for the corresponding author in Editorial Manager on papers submitted after December 6th, 2016. Please ensure that you have an ORCID iD and that it is validated in Editorial Manager. To do this, go to ‘Update my Information’ (in the upper left-hand corner of the main menu), and click on the Fetch/Validate link next to the ORCID field. This will take you to the ORCID site and allow you to create a new iD or authenticate a pre-existing iD in Editorial Manager. Please see the following video for instructions on linking an ORCID iD to your Editorial Manager account: https://www.youtube.com/watch?v=_xcclfuvtxQ.

Additional Editor Comments:

Dear Authors

I am writing to express my gratitude for your submission of “Exploring the Drivers of Risky Sexual Behavior and its social determinants in Nepali Youth: Insights from NDHS 2022” PONE-D-24-00348 to the PLOS ONE journal. I am pleased to inform you that your paper has undergone initial review, and we have received feedback from two reviewers who have provided valuable insights and suggestions for improvement.

Firstly, I would like to commend you on the quality of your work. We found merit in your research, acknowledging its potential significance to the field. However, there are several areas identified where revisions could enhance the clarity, rigor, and impact of your findings.

I have attached the reviewers' comments and suggestions for your consideration. Each comment has been carefully addressed to guide you in addressing specific aspects of your paper. While some revisions may require substantial changes.

I kindly request that you carefully review the feedback provided by the reviewers and consider how the best to incorporate their suggestions into your revised manuscript. Once you have completed your revisions, please resubmit your revised manuscript through our online submission system, along with a detailed response to each reviewer comment explaining how you have addressed their concerns.

We look forward to receiving your revised manuscript.

Shalik Ram Dhital, PhD

Academic Editor

PLOS ONE

Reviewers' comments:

Reviewer's Responses to Questions

**Comments to the Author**

1. Is the manuscript technically sound, and do the data support the conclusions?

Reviewer #1: Yes

Reviewer #2: Partly

Reviewer #3: No

2. Has the statistical analysis been performed appropriately and rigorously? 

Reviewer #1: Yes

Reviewer #2: Yes

Reviewer #3: No

3. Have the authors made all data underlying the findings in their manuscript fully available?

Reviewer #1: Yes

Reviewer #2: Yes

Reviewer #3: Yes

4. Is the manuscript presented in an intelligible fashion and written in standard English?

Reviewer #1: Yes

Reviewer #2: No

Reviewer #3: No

5. Review Comments to the Author

Reviewer #1: This article is well presented and written. I have following comments:

Title: Exploring does not suit for this quantitative study.

Abstract:

The objective verb “delves” need to be replaced with specific action verb.

In methods, authors are explaining determinants as if they were the results.

In the last sentences of Results, there is only one factor mentioned while the sentence starts with “variables that were”

What are the targeted interventions?

Introduction

Introduction needs some specific examples, not only theoretical sentences like “Moreover, the country diverse cultural tapestry encompasses a rich array of traditions, norms, and values that often play a pivotal role in shaping attitudes towards sexuality”

If authors conclude in introduction (the high risk of risky sexual behavior among Nepali youth is influenced by a complex interplay of cultural, educational, economic, gender-related, healthcare-related, and technological factors); why this study is needed. It seems like already known. That is why authors need to be careful on what research has been done and what not, to justify this study.

Discussion:

Are there any limitations for this study?

Conclusion:

What are the vital insights? Need to specify.

Also specify the targeted interventions you can suggest based on findings.

Reviewer #2: The authors have demonstrated significant effort in utilizing NDHS data to offer a nationally representative insight into the risky sexual behavior among youth. However, there are areas in the writing that require refinement. Specifically, language usage can be enhanced, and grammatical errors need attention through proofreading.

Below are some significant points for improvement:

Introduction:

1. This section lacks strength and fails to effectively convey the significance of the research. It is essential to strengthen the introduction by articulating why this research is important and building a compelling narrative. Emphasizing the research gap and illustrating how this study addresses it is crucial. Providing specific data and prevalence rates, if available, to support arguments would strengthen the introduction further.

2. Introduction 3rd para: “ In developing country like Nepal, there is increasing modernization of society and culture especially among the young generation which has brought a positive attitude towards the sexual behavior and other risky behaviors which make them more vulnerable to STIs (11).”This sentence is confusing and long. Please consider shortening it or breaking it down. It is unclear what do you mean by “positive attitude towards sexual behaviors”, and if it is positive why does it “make them more vulnerable”?

3. “Moreover, the country” – change ‘country’ to ‘country’s’?

4. ‘Education and economic status are known to be influential determinants of sexual behavior’- provide reference for this sentence

5. ‘It may also influence decisions related to family planning and contraception’- please avoid using ambiguous pronoun, be more specific what “it” means when beginning a sentence

6. Please use “the” after “despite” – despite the efforts instead of despite effort

Methods

1. Please specify the specific datasets used from the NDHS, such as the IR (Individual Recode) file and MR (Men's Recode) file.

2. Elaborate on the modifications made to variables for this study, such as the wealth index. Provide rationale for why it was modified to only three categories, and extend this explanation to other modified variables.

3. It is advisable to include specific subsections under the methods section. Additionally, consider adding a subsection on variables, clearly outlining the outcome variables with their operational definitions and independent variables. Elaborate on any modifications made to these variables.

4. Specify the variables used for weighting, considering that different datasets may require different weighting variables, particularly for men and women. Describe how this variation was accounted for in the analysis.

5. Explain the rationale for extracting data to SPSS rather than directly to an R file.

6. Revise the statement "A P < 0.05 was used to define the statistical significance" to make it sound more professional and academic.

7. Elaborate on the method used to identify the most important variables for inclusion in the multivariate logistic regression models. Explain why explanatory determinants were initially included one at a time to examine their univariate relationship with the outcome, and how multivariate logistic regression models were subsequently used to identify the most important determinants for each outcome.

8. Reframe the sentences of ethical considerations to clarify that ethical approval was not sought for this study. Explain that the authors used secondary data that had already undergone ethical approval for the original NDHS. Specify how and where the dataset was obtained, emphasizing the use of anonymized data that is publicly available. Describe the approval process from the DHS to gain access to the data.

Results

Revise the following sentence “In terms of provinces, the Madhesh province, Bagmati province, and Lumbini province are among the most highly represented provinces.” Please clarify the province was highly representative for what?

As of PLOS guidelines, the tables should be in-text, right below the text explaining the findings and all cells should be outlined.

Please shorten the descriptive results and highlight the key findings only in the text instead of repeating all the results presented in the table.

Discussion

1. First paragraph: Please summarize the key results—the main takeaway from the results section for a smooth transition to the discussion section. This introductory paragraph of the discussion should guide readers through the results that will be elaborated on in subsequent paragraphs.

2. Improving the discussion: Enhance the discussion by emphasizing how your research addresses research gaps. Highlighting the policy implications is also recommended, particularly as you're working with nationally representative datasets. Stakeholders like the Ministry of Health and Population (MOHP) would find these results valuable, so linking them to policies could be fruitful.

3. Include a limitations section in the discussion. It's essential for every research study to honestly acknowledge limitations, focusing on both the limitations of the researchers and limitations in the data.

Conclusion

This section should reiterate the main takeaway from the results, emphasizing their implications for future actions. Recommendations should be pragmatic and rooted in the findings. Since the study did not directly measure the need for comprehensive education, it's crucial to focus on data-driven decision-making. Thus, the conclusion and recommendations should be closely linked to the results. Additionally, suggesting avenues for future research based on the limitations of this study and identified data gaps could be a key recommendation.

Reviewer #3: Thank you, editor, for sending this manuscript review. This reviewer has following observation.

1. Authors included major background variables and some behavioural variables in the analysis. However, from the policy and programs point of view, important variables are missing. For example, for the outcome variables of recent sexual activity and extra marital activities, whether their partners are living with them or not, use of family planning services in the recent sexual activity, age at their marriage, number of children, some variables related to health seeking practices for sexual and reproductive health. This information are variables in the NDHS 2022 dataset which requires to further dig down rather than restricting analysis traditional variables. Regarding the pre-marital intercourse, there are some behavioural information which authors can look and answers the why questions.

Except this major analytical issue, authors can improve syntax issues in writing and improve readability of the paper.

Authors presented the yes percentage and total n of the row numbers in the same place. Please first put yes % and next column put total from which that row percentage was calculated.

6. PLOS authors have the option to publish the peer review history of their article (what does this mean?). If published, this will include your full peer review and any attached files.

Reviewer #1: **Yes: **Rajendra Karkee

Reviewer #2: No

Reviewer #3: No

---

## [Author Response · Author response to Decision Letter 0]

21 Apr 2024

REVIEWER 1 EVALUATION

This article is well presented and written. I have following comments:

1. Title: Exploring does not suit for this quantitative study.

Author Response: Exploring has been removed from the title as suggested. We had added the word Assessing instead of exploring. 

2. Abstract:

The objective verb “delves” need to be replaced with specific action verb.

Author Response: The verb ‘delves’ has been replaced as suggested. Replace it with word “examine” on page 2, line 23. 

In methods, authors are explaining determinants as if they were the results.

Author Response: We had revised the method section as suggested on page 2, line 26-29.

In the last sentences of Results, there is only one factor mentioned while the sentence starts with “variables that were”.

Author Response: The sentence has been revised as suggested page 2-3, line 39- 41.

What are the targeted interventions?

Author Response: The targeted interventions could be the sexual health education for the youths which has been mentioned in the conclusion section on page 3, line 46-48.

3. Introduction

Introduction needs some specific examples, not only theoretical sentences like “Moreover, the country diverse cultural tapestry encompasses a rich array of traditions, norms, and values that often play a pivotal role in shaping attitudes towards sexuality”

Author Response: The sentence had been removed and we have revised the section at page 4, line 80-85. 

If authors conclude in introduction (the high risk of risky sexual behavior among Nepali youth is influenced by a complex interplay of cultural, educational, economic, gender-related, healthcare-related, and technological factors); why this study is needed. It seems like already known. That is why authors need to be careful on what research has been done and what not, to justify this study.

Author Response: Thank you for the suggestion. We have revised the manuscript based on your suggestions. We had update the manuscript adding the rationale why this study is needed, what research has been done and what not, to justify this study. The revised text is on Page 6, 112-124. 

Discussion:

Are there any limitations for this study?

Author Response: Limitations has been added after the discussion on page 38, line 434-435.

Conclusion:

What are the vital insights? Need to specify.

Author Response: We had specific vital insights in page 39, line 438-450.

Also specify the targeted interventions you can suggest based on findings.

Author Response: The targeted intervention for suggestion has been added in page 39, line 441-450.

REVIEWER 2 EVALUATION

The authors have demonstrated significant effort in utilizing NDHS data to offer a nationally representative insight into the risky sexual behaviour among youth. However, there are areas in the writing that require refinement. Specifically, language usage can be enhanced, and grammatical errors need attention through proofreading.

Below are some significant points for improvement:

Introduction:

1. This section lacks strength and fails to effectively convey the significance of the research. It is essential to strengthen the introduction by articulating why this research is important and building a compelling narrative. Emphasizing the research gap and illustrating how this study addresses it is crucial. Providing specific data and prevalence rates, if available, to support arguments would strengthen the introduction further.

Author Response: Thank you for the suggestion. Based on your suggestions, we had revised the manuscript, adding the significance of the research, research gaps. The revised text is on Page 6, 112-124. We also add the specific data and prevalence rate on Page 5, 85-90.

2. Introduction 3rd para: “ In developing country like Nepal, there is increasing modernization of society and culture especially among the young generation which has brought a positive attitude towards the sexual behavior and other risky behaviors which make them more vulnerable to STIs (11).”This sentence is confusing and long. Please consider shortening it or breaking it down. It is unclear what do you mean by “positive attitude towards sexual behaviors”, and if it is positive why does it “make them more vulnerable”?

Author Response: We have break down the sentence and revised it as suggested on page 4, line 81-85.

3. “Moreover, the country” – change ‘country’ to ‘country’s’?

Author Response: The sentence has been revised to add the data in that section.

4. ‘Education and economic status are known to be influential determinants of sexual behaviour’- provide reference for this sentence

Author Response: Reference has been added.

5. ‘It may also influence decisions related to family planning and contraception’- please avoid using ambiguous pronoun, be more specific what “it” means when beginning a sentence

Author Response: The sentence has been revised.

6. Please use “the” after “despite” – despite the efforts instead of despite effort

Author Response: ‘The’ was added after despite as suggested.

Methods

1. Please specify the specific datasets used from the NDHS, such as the IR (Individual Recode) file and MR (Men's Recode) file.

Author Response: The data set used from the NDHS were NPMR82SV and NPIR82SV. The name of specific datasets is mentioned in Page 8 line 152-153.

2. Elaborate on the modifications made to variables for this study, such as the wealth index. Provide rationale for why it was modified to only three categories, and extend this explanation to other modified variables.

Author Response: The wealth index variable was modified from five categories to three categories for easy interpretation of the data, ease to communicate and data validation. We had added this information in Page 9, 173-175. 

3. It is advisable to include specific subsections under the methods section. Additionally, consider adding a subsection on variables, clearly outlining the outcome variables with their operational definitions and independent variables. Elaborate on any modifications made to these variables.

Author Response: Based on we had added the subsection, study design, study settings, sample size, study variables, statistical analysis and ethics statement. 

4. Specify the variables used for weighting, considering that different datasets may require different weighting variables, particularly for men and women. Describe how this variation was accounted for in the analysis.

Author Response: We have added this information in Page 9, 181-187.

5. Explain the rationale for extracting data to SPSS rather than directly to an R file.

Author Response: There is no specific rationale for extracting data to SPSS rather than directly to an R file. The NDHS datasets are available in SPSS. The research team is expert in using R program and the program had advantages of complex survey analysis.

6. Revise the statement "A P < 0.05 was used to define the statistical significance" to make it sound more professional and academic.

Author Response: The sentence has been revised on page 9, line 193-194.

7. Elaborate on the method used to identify the most important variables for inclusion in the multivariate logistic regression models. Explain why explanatory determinants were initially included one at a time to examine their univariate relationship with the outcome, and how multivariate logistic regression models were subsequently used to identify the most important determinants for each outcome.

Author Response: Based on the reviewers’ comments, we had revised the manuscript and added the information on using the multivariate logistic regression. The revised text is on Page 10, line 199-200.

8. Reframe the sentences of ethical considerations to clarify that ethical approval was not sought for this study. Explain that the authors used secondary data that had already undergone ethical approval for the original NDHS. Specify how and where the dataset was obtained, emphasizing the use of anonymized data that is publicly available. Describe the approval process from the DHS to gain access to the data.

Author Response: Thank you for the suggestion. We have specified how the data set was obtained and approval process on page 11, line 224- 227. 

Results

1. Revise the following sentence “In terms of provinces, the Madhesh province, Bagmati province, and Lumbini province are among the most highly represented provinces.” Please clarify the province was highly representative for what?

Author Response: The sentence has been revised on page 11-12, line 236-237. 

2. As of PLOS guidelines, the tables should be in-text, right below the text explaining the findings and all cells should be outlined.

Author Response: The tables have been inserted in- text below the text.

3. Please shorten the descriptive results and highlight the key findings only in the text instead of repeating all the results presented in the table.

Author Response: The descriptive results have been shortened as suggested. 

Discussion

1. First paragraph: Please summarize the key results—the main takeaway from the results section for a smooth transition to the discussion section. This introductory paragraph of the discussion should guide readers through the results that will be elaborated on in subsequent paragraphs.

Author Response: Based on your suggestions, we have revised discussion and update text in the revised manuscript. The first paragraph has been revised on page 34, line 327- 331. 

2. Improving the discussion: Enhance the discussion by emphasizing how your research addresses research gaps. Highlighting the policy implications is also recommended, particularly as you're working with nationally representative datasets. Stakeholders like the Ministry of Health and Population (MOHP) would find these results valuable, so linking them to policies could be fruitful.

Author Response: Thank you for suggestion. Though there are not lots of such policies that could be directly linked to but we have mentioned about the laws regarding the protection of minors. However we had add the research gaps and recommendations for MOHP in page 38, line 427-432.

3. Include a limitations section in the discussion. It's essential for every research study to honestly acknowledge limitations, focusing on both the limitations of the researchers and limitations in the data.

Author Response: Limitations has been added on page 38, line 434- 435. 

Conclusion

This section should reiterate the main takeaway from the results, emphasizing their implications for future actions. Recommendations should be pragmatic and rooted in the findings. Since the study did not directly measure the need for comprehensive education, it's crucial to focus on data-driven decision-making. Thus, the conclusion and recommendations should be closely linked to the results. Additionally, suggesting avenues for future research based on the limitations of this study and identified data gaps could be a key recommendation.

Author Response: We have revised the conclusion and recommendations based on the reviewers feedback. Now the conclusion and recommendations is linked to the results section. The revised text is page 39, line 438-441. 

REVIEWER 3 EVALUATION

Thank you, editor, for sending this manuscript review. This reviewer has following observation.

1. Authors included major background variables and some behavioural variables in the analysis. However, from the policy and programs point of view, important variables are missing. For example, for the outcome variables of recent sexual activity and extra marital activities, whether their partners are living with them or not, use of family planning services in the recent sexual activity, age at their marriage, number of children, some variables related to health seeking practices for sexual and reproductive health. This information are variables in the NDHS 2022 dataset which requires to further dig down rather than restricting analysis traditional variables. Regarding the pre-marital intercourse, there are some behavioural information which authors can look and answers the why questions.

Except this major analytical issue, authors can improve syntax issues in writing and improve readability of the paper.

Authors Response: Thank you for your insightful comments. We agree that variables such as partner living arrangements, family planning use, age at marriage, number of children, and health-seeking practices for sexual and reproductive health are crucial from a policy and program perspective. However, during our analysis, we found that the sample size may not have been sufficient to thoroughly explore all the variables, for example condom use in last sex among youths, as the data was limited to do multivariate logistic regression.

Thank you for highlighting these important areas for improvement, and we revised our manuscript incorporating these suggestions.

2. Authors presented the yes percentage and total n of the row numbers in the same place. Please first put yes % and next column put total from which that row percentage was calculated.

Authors Response: Based on the feedback from the reviewers, we had revised the table. The revised table is added to the revised manuscript.

---

## [Decision Letter · Decision Letter 1]

15 May 2024

PONE-D-24-00348R1Assesssing the Drivers of Risky Sexual Behavior among youths and its social determinants in NepalPLOS ONE

Dear Dr. Kakchapati,

Thank you for submitting your manuscript to PLOS ONE. After careful consideration, we feel that it has merit but does not fully meet PLOS ONE’s publication criteria as it currently stands. Therefore, we invite you to submit a revised version of the manuscript that addresses the points raised during the review process.

We look forward to receiving your revised manuscript.

Kind regards,

Shalik Ram Dhital, PhD

Academic Editor

PLOS ONE

Journal Requirements:

*Comments from PLOS Editorial Office:*

1) We are concerned that your definition of risky sexual behaviour is inconsistent with the definitions used elsewhere in the scientific literature. Although there is no single definitive list of risky sexual behaviours, most research includes unprotected sex, sex under the influence of drugs/alcohol, and multiple sexual partners. 

In contrast, you list premarital sex, recent sex, and multiple sexual partners, but do not examine unprotected sex. Your provide citations, but neither of these references define premarital sex or recent sex as risky sexual behaviours. Azene et al. (2022) define risky sexual behaviour as “having sex with multiple sexual partners, early initiation of sexual intercourse under the age of 18, and not using or inconsistent use of a condom.” Keto et al. (2020) mention “having multiple sexual partners, sexual intercourse with commercial sex workers, unprotected sexual intercourse, coerced sexual intercourse and sexual intercourse for reward.” 

We request that you analyse outcome variables that are typically agreed upon as risky sexual behaviours (e.g., unprotected sex), and do not include premarital sex or recent sex as measures of risky sexual behaviour without thorough justification. Furthermore, your definition of pre-marital sex is restricted to “never-married youths who had ever had sexual intercourse”. This excludes respondents who were married at the time of the survey but had engaged in pre-marital sex. In addition, you define recent sexual activity as “having sexual intercourse in the last 12 months with a person who neither was their wife nor lived with them,” which seems to be measuring the same construct as premarital sex, only restricted to men and to the year prior to the survey.

2) Although you note that the “cross-sectional nature of the data collection limits the study from establishing a causal relationship,” you offer causal explanations for the correlations between the predictor and outcome variables.  For example, the association between tobacco use and the outcome variables is explained as "tobacco products contain nicotine which impairs cognitive function of brain and lowers the decision-making process", with no evidence for this assertion, and without noting that both tobacco use and risky sexual behaviour may be the outcome of a third factor, such as impulsivity. We request that you limit your discussion and conclusions to observations that can reasonably be inferred from the data.

Additional Editor Comments:

Dear Authors,

Your revision version of this manuscript has improvement. However, There are still lacking few issues commented by reviewers 2 and 3. I have provided my comments in track change manuscript attached. Please go through my comments and also with reviewers comments.

I am looking forward to receiving your revised version.

Kind regards

Shalik Ram Dhital, PhD

Academic Editor

PLOS ONE

Reviewers' comments:

Reviewer's Responses to Questions

**Comments to the Author**

1. If the authors have adequately addressed your comments raised in a previous round of review and you feel that this manuscript is now acceptable for publication, you may indicate that here to bypass the “Comments to the Author” section, enter your conflict of interest statement in the “Confidential to Editor” section, and submit your "Accept" recommendation.

Reviewer #1: All comments have been addressed

Reviewer #2: (No Response)

Reviewer #3: All comments have been addressed

2. Is the manuscript technically sound, and do the data support the conclusions?

Reviewer #1: Yes

Reviewer #2: Partly

Reviewer #3: Yes

3. Has the statistical analysis been performed appropriately and rigorously? 

Reviewer #1: Yes

Reviewer #2: Yes

Reviewer #3: Yes

4. Have the authors made all data underlying the findings in their manuscript fully available?

Reviewer #1: Yes

Reviewer #2: Yes

Reviewer #3: Yes

5. Is the manuscript presented in an intelligible fashion and written in standard English?

Reviewer #1: Yes

Reviewer #2: No

Reviewer #3: Yes

6. Review Comments to the Author

**Reviewer #1: **The authors have addressed my comments and has revised the manuscripts. It is satisfactory and I recommend publication of this article.

**Reviewer #2:** Reviewer's Comments:

The authors have made significant improvements in addressing previous comments. However, the discussion section still requires further improvement. It currently lacks depth, merely restating results without critical analysis or comparison with existing literature. Moreover, it fails to adequately address the policy implications of the findings.

Specifically, the discussion should go beyond simply presenting results and delve into comparisons with other studies, highlighting similarities, differences, and contributing factors. Additionally, it should explore the implications of the findings for policy formulation and intervention strategies.

The literature review appears insufficient, with a lack of citations to support the discussion points. It is recommended to bolster the discussion with relevant references, particularly when mentioning legal frameworks or societal norms.

Please ensure to integrate potential recommendations or policy implications into each paragraph of the discussion. While it doesn't need to be extensive, a few lines with appropriate references are essential. Consider what the existing literature suggests and what interventions have been successful in Nepal. If local examples are lacking, draw upon effective interventions from global literature, particularly from LMICs. This addition will enrich the discussion and provide actionable insights for policymakers and practitioners.

For example,

Lines 264-267 "Although we have existing laws intended at protecting minors from engaging in sexual activities but still there are prevalence of sexual activity which needs to be addressed through awareness programs targeting

adolescents and youths." When referencing the law, please ensure to provide a solid reference. You can cite various legal acts in Nepal by referring to the Nepal Law Commission for accurate information.

Please provide references for the sentences in lines 292-296, which discuss patriarchy and gender inequality. For example, "Females are expected to maintain their virginity till marriage" and "Females are at higher risk of experiencing sexual health consequences..." These statements should be supported by relevant studies or literature on the topic.

Please provide references for the paragraph from lines 298 to 304, which discusses migration and sexual health. Studies conducted in regions such as Gandaki, Karnali, and Sudurpashchim would be pertinent to support the assertions made in this paragraph. You can refer to higher migration rates and risky sexual behaviors, higher prevalence of STI/HIV in certain regions etc.

Lines 305-308, please provide references

For lines 309 to 314, if literature from Nepal is unavailable, please search for relevant global literature from LMICs. Look for studies that have similar findings regarding the topic discussed. If no such studies are available, explicitly state that these findings are unique to Nepal. Additionally, consider including suggestions on how to identify the specific group mentioned in lines 309 to 314 for targeted interventions, such as awareness programs. This could involve utilizing demographic data, community outreach, or other methods.

Limitations section lines 351-352

The authors have stated that "this study has some limitations," but have only included one sentence on the study design. There should be more limitations. Just two lines of limitation are too simplistic. How about the use of different variables? How about limitations of not including certain variables that would have been interesting? Also, you may add that despite the limitations, this study has its strengths too, and highlight the strengths as well to provide a more balanced perspective.

In terms of specific revisions on typos that need thorough proofreading:

Abstract

Line 25: There's a typo, an "e" after "prevalence of".

Introduction:

Line 67: Put all references inside one parenthesis, separated by commas or hyphens.

Line 72: Remove the redundant "in" after "NDHS" and "revealed".

Lines 67-71: Break down the lengthy sentence for better readability.

Line 96: Remove one of the duplicate "that" to avoid redundancy.

Line 121: Change "include" to "included".

Line 129: Unbold "partners".

Line 137: Change "used" to "use of media".

Line 142: Remove the closing parenthesis after "communication and data limitations".

Line 174: Remove the repetitive sentence about statistical analysis as it has already been explained earlier.

Line 187: Capitalize "janajati" to "Janajati".

Line 191: Remove the extra comma after "having six or more members".

Table 1: Keep the style consistent for age ("51%") versus occupation ("5.24"). Since the labeling specifies "%" at the top, it's unnecessary to repeat it after each number.

Table 2: Format the table appropriately.

**Reviewer #3: **Reviewer thanks authors for addressing comments. However, discussion section has still lacked articulation of findings with existing policies and plans, programs and strategies, and services. There are several tyos in the manuscript which needs careful editing. In addition, in the table if we write % in the tables, it is unnecessary to write % in every row, also no need to write province for each province in the table. These are very minor but needs careful editing and formatting of the texts, references, etc.

7. PLOS authors have the option to publish the peer review history of their article (what does this mean?). If published, this will include your full peer review and any attached files.

Reviewer #1: **Yes: **Rajendra Karkee

Reviewer #2: No

Reviewer #3: No

---

## [Author Response · Author response to Decision Letter 1]

28 Jun 2024

Dear Academic Editor 

Dr. Shalik Ram Dhital

PLOS ONE

We would like to thank you and the reviewers for the thorough and constructive feedback on our manuscript titled “Assessing the drivers of sexual behaviour among youths and its social determinants in Nepal” [PONE-D-24-00348R1]. We have carefully considered all the comments from the reviewers and suggestions provided, and we believe that the revisions have significantly improved the quality of our manuscript. Below, we provide a detailed response to each point raised.

Comments from PLOS Editorial Office:

1) We are concerned that your definition of risky sexual behaviour is inconsistent with the definitions used elsewhere in the scientific literature. Although there is no single definitive list of risky sexual behaviours, most research includes unprotected sex, sex under the influence of drugs/alcohol, and multiple sexual partners. In contrast, you list premarital sex, recent sex, and multiple sexual partners, but do not examine unprotected sex. Your provide citations, but neither of these references define premarital sex or recent sex as risky sexual behaviours. Azene et al. (2022) define risky sexual behaviour as “having sex with multiple sexual partners, early initiation of sexual intercourse under the age of 18, and not using or inconsistent use of a condom.” Keto et al. (2020) mention “having multiple sexual partners, sexual intercourse with commercial sex workers, unprotected sexual intercourse, coerced sexual intercourse and sexual intercourse for reward.” We request that you analyse outcome variables that are typically agreed upon as risky sexual behaviours (e.g., unprotected sex), and do not include premarital sex or recent sex as measures of risky sexual behaviour without thorough justification. Furthermore, your definition of pre-marital sex is restricted to “never-married youths who had ever had sexual intercourse”. This excludes respondents who were married at the time of the survey but had engaged in pre-marital sex. In addition, you define recent sexual activity as “having sexual intercourse in the last 12 months with a person who neither was their wife nor lived with them,” which seems to be measuring the same construct as premarital sex, only restricted to men and to the year prior to the survey.

Authors Response: We appreciate the reviewers' thorough feedback and the opportunity to improve our manuscript. We understand the importance of aligning our definitions of risky sexual behavior with those commonly accepted in scientific literature. After careful consideration of your comments, we have made the following adjustments to our manuscript:

We agree with the reviewers' comments regarding the definition of risky sexual behaviors. Based on the reviewers’ suggestions, we agreed that our study doesn’t measure and match the risky sexual behaviors. So, to address this, we have revised the terminology from "risky sexual behaviors" to "sexual behaviors." Now, we will use sexual behaviors in the revised manuscript instead of risky sexual behavior. This change reflects a broader and more inclusive understanding of the sexual behaviors we are studying. The sexual behaviors definition include premarital sex, multiple sexual partners and recent sexual activity. While we acknowledge that premarital sex and recent sexual activity are not typically classified as "risky sexual behaviors" in the broader literature, we believe they are important to study within our specific cultural context. These behaviors can have significant social and health implications, and therefore warrant inclusion in our analysis as "sexual behaviors." We made the necessary changes to the overall manuscript. We believe these revisions address the reviewers' concerns and improve the clarity and rigor of our manuscript. Thank you for your valuable feedback, which has significantly improved our study.

2) Although you note that the “cross-sectional nature of the data collection limits the study from establishing a causal relationship,” you offer causal explanations for the correlations between the predictor and outcome variables. For example, the association between tobacco use and the outcome variables is explained as "tobacco products contain nicotine which impairs cognitive function of brain and lowers the decision-making process", with no evidence for this assertion, and without noting that both tobacco use and risky sexual behaviour may be the outcome of a third factor, such as impulsivity. We request that you limit your discussion and conclusions to observations that can reasonably be inferred from the data.

Response: We appreciate the reviewers' valuable feedback and the opportunity to improve our manuscript. We understand the importance of appropriately interpreting the findings from cross-sectional data and the necessity of avoiding causal assertions without strong evidence. To address the reviewers' concerns, we have made the following adjustments to our manuscript:

We have emphasized the limitations of cross-sectional data in the discussion section, specifically noting that such data only allows for the identification of associations rather than causal relationships. This includes acknowledging that any observed associations between predictor and outcome variables cannot be used to infer causation. We have revised our discussion to ensure that all interpretations of the data are appropriately cautious and framed within the context of association rather than causation. We had addressed all the comments in the revised manuscript. 

Reviewer #2: Reviewer's Comments:

The authors have made significant improvements in addressing previous comments. However, the discussion section still requires further improvement. It currently lacks depth, merely restating results without critical analysis or comparison with existing literature. Moreover, it fails to adequately address the policy implications of the findings. Specifically, the discussion should go beyond simply presenting results and delve into comparisons with other studies, highlighting similarities, differences, and contributing factors. Additionally, it should explore the implications of the findings for policy formulation and intervention strategies. The literature review appears insufficient, with a lack of citations to support the discussion points. It is recommended to bolster the discussion with relevant references, particularly when mentioning legal frameworks or societal norms. Please ensure to integrate potential recommendations or policy implications into each paragraph of the discussion. While it doesn't need to be extensive, a few lines with appropriate references are essential. Consider what the existing literature suggests and what interventions have been successful in Nepal. If local examples are lacking, draw upon effective interventions from global literature, particularly from LMICs. This addition will enrich the discussion and provide actionable insights for policymakers and practitioners.

Author Response: Thank you for your detailed and constructive feedback. We appreciate your efforts to help improve our manuscript and understand the need for a more in-depth discussion and are committed to enhancing this section to meet the required standards. Based on your suggestions, we have enriched the discussion by providing a more critical analysis of our findings. Specifically, we compare our results with those of other relevant studies, both locally and globally, to highlight similarities, differences, and contributing factors. And we also examine how our findings align or contrast with existing research on risky sexual behavior and its social determinants, drawing on studies from other LMICs to provide a broader context. To address the policy implications of our findings, we integrate potential recommendations and intervention strategies in the discussion section. 

Reviewers comments

Lines 264-267 "Although we have existing laws intended at protecting minors from engaging in sexual activities but still there are prevalence of sexual activity which needs to be addressed through awareness programs targeting adolescents and youths." When referencing the law, please ensure to provide a solid reference. You can cite various legal acts in Nepal by referring to the Nepal Law Commission for accurate information.

Author Response: Thank you for the suggestion. The existing laws has been cited as suggested in page 24 line 283.

Please provide references for the sentences in lines 292-296, which discuss patriarchy and gender inequality. For example, "Females are expected to maintain their virginity till marriage" and "Females are at higher risk of experiencing sexual health consequences..." These statements should be supported by relevant studies or literature on the topic.

Author Response: Relevant literatures has been added which supports the statement in Page 25, lines 303 and 306.

Please provide references for the paragraph from lines 298 to 304, which discusses migration and sexual health. Studies conducted in regions such as Gandaki, Karnali, and Sudurpashchim would be pertinent to support the assertions made in this paragraph. You can refer to higher migration rates and risky sexual behaviors, higher prevalence of STI/HIV in certain regions etc.

Author Response: Thank you for the comments and suggestions. Related references which has shown prevalence of HIV/ STD among migrant workers has been added as suggested in Page 25 line 313 and 315.

Lines 305-308, please provide references

Author Response: Reference has been added in line 395.

For lines 309 to 314, if literature from Nepal is unavailable, please search for relevant global literature from LMICs. Look for studies that have similar findings regarding the topic discussed. If no such studies are available, explicitly state that these findings are unique to Nepal. Additionally, consider including suggestions on how to identify the specific group mentioned in lines 309 to 314 for targeted interventions, such as awareness programs. This could involve utilizing demographic data, community outreach, or other methods.

Author Response: Thank you for comments and suggestions. We have added the references based on your suggestions. The targeted interventions have been added as suggested in Page 26 lines 325-330. .

Limitations section lines 351-352

The authors have stated that "this study has some limitations," but have only included one sentence on the study design. There should be more limitations. Just two lines of limitation are too simplistic. How about the use of different variables? How about limitations of not including certain variables that would have been interesting? Also, you may add that despite the limitations, this study has its strengths too, and highlight the strengths as well to provide a more balanced perspective.

Author Response: Thank you for your suggestions. Based on your feedback, we had added some limitations for the study. Additional limitations and strengths for the study have been added in Page 27-28 lines 360-365

Reviewer #3: Reviewer thanks authors for addressing comments. However, discussion section has still lacked articulation of findings with existing policies and plans, programs and strategies, and services. 

Author Response: 

There are several tyos in the manuscript which needs careful editing. In addition, in the table if we write % in the tables, it is unnecessary to write % in every row, also no need to write province for each province in the table. These are very minor but needs careful editing and formatting of the texts, references, etc.

Author Response: Typos error has been removed from Table 1.

---

## [Decision Letter · Decision Letter 2]

19 Jul 2024

PONE-D-24-00348R2Assesssing the Drivers of Sexual Behavior and its social determinants in Nepali Youth: Insights from NDHS 2022PLOS ONE

Dear Dr. Kakchapati,

Thank you for addressing most of the feedback provided by reviewers and editor. There are few minor issue you need to resolve before processing your paper. Please go through track change paper with my comments and address these issue and resubmit your final version two copies 1. track change and 2. a clean copy.

Looking forward to receiving your revised copy

Kind regards

Shalik Dhital, PhD

Academic Editor

PLOS ONE

Journal Requirements:

*Comments from PLOS Editorial Office:*

Thank you for taking our concerns into account when revising your manuscript. However, some of the issues remain. For example, you still state: "the use of any tobacco products, either cigarettes or electronic cigarettes (vape), or chewing tobacco can impair judgment leading to riskier decision-making and practicing risky activities like engaging in sexual behavior which they might avoid if sober." Please can you carefully revise your manuscript to remove all causal explanations for the correlations between the predictor and outcome variables, particularly where there is no evidence to support such causal relationships.

Reviewers' comments:

Reviewer's Responses to Questions

**Comments to the Author**

1. If the authors have adequately addressed your comments raised in a previous round of review and you feel that this manuscript is now acceptable for publication, you may indicate that here to bypass the “Comments to the Author” section, enter your conflict of interest statement in the “Confidential to Editor” section, and submit your "Accept" recommendation.

Reviewer #2: All comments have been addressed

Reviewer #3: All comments have been addressed

2. Is the manuscript technically sound, and do the data support the conclusions?

Reviewer #2: Yes

Reviewer #3: Yes

3. Has the statistical analysis been performed appropriately and rigorously? 

Reviewer #2: Yes

Reviewer #3: Yes

4. Have the authors made all data underlying the findings in their manuscript fully available?

Reviewer #2: Yes

Reviewer #3: Yes

5. Is the manuscript presented in an intelligible fashion and written in standard English?

Reviewer #2: Yes

Reviewer #3: Yes

6. Review Comments to the Author

Reviewer #2: Thank you for revising the manuscript and addressing all the comments. Most of the major comments are addressed and the current version of the manuscript is much better.

There are just a few final minor comments as follows:

Introduction

1. The authors are still including "premarital sex" as risky sexual behaviors at some places – please consider revising and removing “premarital sex” from risky sexual behavior. The authors must understand that premarital sex among two consenting adults is not risky, as long as it is safe and as long as they are aware. Sex among under aged people can still be risky even if they are married. Using a blanket term of “premarital sex” for risky sexual behavior creates stigma that is not helpful for public health interventions promoting healthy sexual behaviors among young people. The stigma surrounding premarital sex is also one of the reasons why early marriages are still prevalent in Nepal.

The risky behaviors related to sex goes beyond marital status. While the authors have been more careful with the terms linking “risky sexual behaviors” with “premarital sex”, it would be helpful for authors to review the manuscript carefully again and balance the tone wherever needed.

Therefore, please remove “premarital sex” from Line 47 of the revised file where it states

“Risky sexual behavior can be described as any sexual activity which has greater risk of having

sexually transmitted infections (STIs) and unintended pregnancy [1,2].It also includes having sexual relation with more than one partner, having intercourse at an early age or under 18 years, PREMARITAL SEX, engaging in sexual activity under the influence of drugs or alcohol, and not using condom [2,3]”

2. Line 63-67 - “This shift led to a more open attitude towards sexual behaviors and other risk behaviors, develop positive attitude in promoting discussions around sexual health, also increases their vulnerability to STIs/HIV [11], for example a previous study among the adolescents showed that 59% of the teenagers had just a moderate understanding of sexual and reproductive health [11,12].”

Please consider breaking the following sentence. It is too long and the message is unclear as it includes both positive and negative messages. It can instead be revised as the following (or something similar)

“This shift led to a more open attitude towards sexual behaviors and other risk behaviors increases their vulnerability to STIs/HIV [11], for example a previous study among the adolescents showed that 59% of the teenagers had just a moderate understanding of sexual and reproductive health. However, it has also enabled positive attitude in promoting discussions around sexual health which provides an opportunity for more effective intervention [11,12].”

3. Lines 67-72 -“Moreover, the Nepal Demographic and Health Survey (NDHS) in 2016 revealed that 50% of young women and 42% of young men aged 15-24 have ever had sex. Among never married youths, about one- fourth of men have ever had 70 sex while only smaller proportion of young women ever had sex. Three percent of the men had two or more sexual partners in the past twelve months with 9% reporting to have sexual intercourse with persons who were neither their wife nor in the relationship [13].”

This section can go up preceding the above section. Consider switching this section with lines 63-67. It will improve the flow of the paragraph.

4. Ethics statement - Please specify that “primary” data was approved by NHRC. The “secondary data” was accessed through the DHS program.

5. Conclusion - Lines 366 to 375 serves as a paragraph on recommendation instead of conclusion. Therefore, please move this section above conclusion as the last paragraph of the main discussion section.

For the conclusion section - make it shorter, add a few lines reminding the readers of the key take away findings and just a line of two on recommendation grounded in the findings.

Reviewer #3: Thanks for the revised version. Authors made the suggested changes in the Table 1 only. But there is need of correction in the second column of all other tables.

7. PLOS authors have the option to publish the peer review history of their article (what does this mean?). If published, this will include your full peer review and any attached files.

Reviewer #2: No

Reviewer #3: No

---

## [Author Response · Author response to Decision Letter 2]

16 Aug 2024

Dear Academic Editor 

Dr. Shalik Ram Dhital

PLOS ONE

We would like to thank you and the reviewers for the thorough and constructive feedback on our manuscript titled “Assessing the Drivers of Risky Sexual Behaviour and its social determinants in Nepali Youth: Insights from NDHS 2022” [PONE-D-24-00348R1]. We have carefully considered all the comments from the reviewers and suggestions provided, and we believe that the revisions have significantly improved the quality of our manuscript. Below, we provide a detailed response to each point raised.

Comments from PLOS Editorial Office:

Introduction

1. The authors are still including "premarital sex" as risky sexual behaviors at some places – please consider revising and removing “premarital sex” from risky sexual behavior. The authors must understand that premarital sex among two consenting adults is not risky, as long as it is safe and as long as they are aware. Sex among under aged people can still be risky even if they are married. Using a blanket term of “premarital sex” for risky sexual behavior creates stigma that is not helpful for public health interventions promoting healthy sexual behaviors among young people. The stigma surrounding premarital sex is also one of the reasons why early marriages are still prevalent in Nepal.

The risky behaviors related to sex goes beyond marital status. While the authors have been more careful with the terms linking “risky sexual behaviors” with “premarital sex”, it would be helpful for authors to review the manuscript carefully again and balance the tone wherever needed.

Therefore, please remove “premarital sex” from Line 47 of the revised file where it states

“Risky sexual behavior can be described as any sexual activity which has greater risk of having sexually transmitted infections (STIs) and unintended pregnancy [1,2].It also includes having sexual relation with more than one partner, having intercourse at an early age or under 18 years, PREMARITAL SEX, engaging in sexual activity under the influence of drugs or alcohol, and not using condom [2,3]”

Response: Thank you for your comments and suggestions. We understand the importance of aligning our definitions of sexual behavior and based on your comments, the definition has been revised. Now our study doesn’t include premarital sex in the definition of risky sexual behavior. The text is revised in the updated manuscript as follows: 

"Certain sexual activities may increase the likelihood of sexually transmitted infections (STIs) and unintended pregnancy [1,2] including having multiple sexual partners, participating in sexual activity under the influence of substances, and not using a condom [2,3]." The revised text is updated in Page 3, line 46-51. 

2. Line 63-67 - “This shift led to a more open attitude towards sexual behaviors and other risk behaviors, develop positive attitude in promoting discussions around sexual health, also increases their vulnerability to STIs/HIV [11], for example a previous study among the adolescents showed that 59% of the teenagers had just a moderate understanding of sexual and reproductive health [11,12].”

Please consider breaking the following sentence. It is too long and the message is unclear as it includes both positive and negative messages. It can instead be revised as the following (or something similar)

“This shift led to a more open attitude towards sexual behaviors and other risk behaviors increases their vulnerability to STIs/HIV [11], for example a previous study among the adolescents showed that 59% of the teenagers had just a moderate understanding of sexual and reproductive health. However, it has also enabled positive attitude in promoting discussions around sexual health which provides an opportunity for more effective intervention [11,12].”

Response: Thank you for the feedback and suggestions. We appreciate the reviewers' valuable feedback and the opportunity to improve our manuscript. We agreed with reviewers’ comments and add the reviewer statement in the revised manuscript. Please see Page 4, line 69-75

3. Lines 67-72 -“Moreover, the Nepal Demographic and Health Survey (NDHS) in 2016 revealed that 50% of young women and 42% of young men aged 15-24 have ever had sex. Among never married youths, about one- fourth of men have ever had 70 sex while only smaller proportion of young women ever had sex. Three percent of the men had two or more sexual partners in the past twelve months with 9% reporting to have sexual intercourse with persons who were neither their wife nor in the relationship [13].” This section can go up preceding the above section. Consider switching this section with lines 63-67. It will improve the flow of the paragraph.

Response: Thank you for the suggestions. We acknowledge the reviewers’ comments and move the section above in the revised manuscript. We also added the latest NDHS data of 2022 in the revised manuscript. Please see Page 4, line 60-65

4. Ethics statement - Please specify that “primary” data was approved by NHRC. The “secondary data” was accessed through the DHS program.

Response: The ethics statement has been revised and updated as suggested.

5. Conclusion - Lines 366 to 375 serves as a paragraph on recommendation instead of conclusion. Therefore, please move this section above conclusion as the last paragraph of the main discussion section. For the conclusion section - make it shorter, add a few lines reminding the readers of the key take away findings and just a line of two on recommendation grounded in the findings.

Response: Thank you for your comments and suggestions. We had make the conclusion shorter and revised the recommendations at Page 24-25, 365-370.

Reviewer #3: Thanks for the revised version. Authors made the suggested changes in the Table 1 only. But there is need of correction in the second column of all other tables.

Response: Thank you for the suggestion. All the tables have been corrected and revised as suggested.

---

## [Editor Report · Decision Letter 3]

28 Aug 2024

PONE-D-24-00348R3Assessing the Drivers of Risky Sexual Behaviour and its social determinants in Nepali Youth: Insights from NDHS 2022PLOS ONE

Dear Dr Kakchapati

Thank you for addressing reviewers comments. However, There is still minor revisions require in the manuscript. Therefore, We strongly suggest you thoroughly copyedit your manuscript for language usage, spelling, punctuation, and grammar. If you do

not know anyone who can help you do this, you may wish to consider employing a professional scientific editing service. 

Please submit your revised manuscript by Oct 12 2024 11:59PM. If you will need more time than this to complete your revisions, please reply to this message or contact the journal office at plosone@plos.org. Please include the following items when submitting your revised manuscript:A rebuttal letter that responds to each point raised by the academic editor and reviewer(s). You should upload this letter as a separate file labeled 'Response to Reviewers'.A marked-up copy of your manuscript that highlights changes made to the original version. You should upload this as a separate file labeled 'Revised Manuscript with Track Changes'.An unmarked version of your revised paper without tracked changes. You should upload this as a separate file labeled 'Manuscript'.Please review your reference list to ensure that it is complete and correct. If you have cited papers that have been retracted, please include the rationale for doing so in the manuscript text, or remove these references and replace them with relevant current references. Any changes to the reference list should be mentioned in the rebuttal letter that accompanies your revised manuscript. If you need to cite a retracted article, indicate the article’s retracted status in the References list and also include a citation and full reference for the retraction notice.

While revising your submission, please upload your figure files to the Preflight Analysis and Conversion Engine (PACE) digital diagnostic tool, https://pacev2.apexcovantage.com/. PACE helps ensure that figures meet PLOS requirements. To use PACE, you must first register as a user. Registration is free. Then, login and navigate to the UPLOAD tab, where you will find detailed instructions on how to use the tool. If you encounter any issues or have any questions when using PACE, please email PLOS at figures@plos.org. Please note that Supporting Information files do not need this step. Looking forward to receiving your revised paper with language review.

Kind regards

Shalik Ram Dhital, PhD

Academic Editor

PLOS ONE

---

## [Author Response · Author response to Decision Letter 3]

3 Oct 2024

To,

The Editor,

PLOS ONE

Subject: Response to the query from reviewers

Dear Editor,

Thank you for the evaluation from the reviewers on manuscript entitled on “Assessing the Drivers of Sexual Behavior and its social determinants in Nepali Youths”.

We have responded all of the queries and comments of the reviewers. Please see below for our response to each of the comments/queries. We have also shared the revised manuscript in track changes for your review and approval. 

Thank you very much for reviewing our study. If you need further clarification on this, please do not hesitate to contact us.

Yours sincerely,

Dr. Sampurna Kakchapati

Response to reviewer 

This is still not necessary as you have explained it in Method section. Please delete this last part

Author Response: We have checked this information and compare with NDHS 2022, it was fine

Please check thoroughly language of the whole paper and format final manuscript according to author guideline. Please read carefully the paper.

Author Response: We have checked thoroughly language of the whole paper and format final manuscript according to author guideline. 

Some of the information is similar with your study as you used NDHS 2022 dataset. Please check and compare with your results. 

Author Response: We have checked this information and compare with NDHS 2022, it was fine. Please see Page 4, lines 69-74. 

Which study ? Where this has been done? Please specify.

Author Response: The information of the study is added in the revised manuscript added in page 5, lines 81-83

Please always use full form at first place of the paper. 

Author Response: Yes, we have use full form in the first place of the manuscript. Please check page Page 4, line 55 and page 4, line 59

Write in full form as this is first time you are referring NHRC.

Author Response: Thank you for suggestions. We had use full form of NHRC at Page 9, line 182.

---

## [Editor Report · Decision Letter 4]

16 Oct 2024

PONE-D-24-00348R4Assessing the Drivers of Sexual Behavior and its social determinants in Nepali YouthsPLOS ONE

Dear Dr. Kakchapati,

Thank you for sending this paper with the necessary modifications. However, there are still some minor errors, so I made track changes throughout the entire document. I also reviewed the language, as there were errors in multiple places. Please go through the revised paper using track changes to address all minor comments and language corrections. After that, please upload both a track change copy and a clean copy when you submit. Additionally, please ensure that the information about data availability criteria, funding sources, and competing interests is included in the paper. Please follow PLOS ONE Author Guidelines. For any queries please contact to plosone@plos.org

With Kind Regards

Shalik Ram Dhital, PhD

Academic Editor

PLOS ONE

---

## [Author Response · Author response to Decision Letter 4]

3 Nov 2024

Dear Editor

Thank you for your feedback and the detailed track changes on our manuscript. We appreciate your efforts in reviewing the paper and addressing the necessary modifications. We have carefully gone through all your comments and suggestions, making the appropriate revisions to the document. Attached, you will find both the track changes copy and a clean version of the manuscript for your review.

If you have any further queries or require additional adjustments, please do not hesitate to reach out.

Thank you once again for your guidance.

Sampurna

---

## [Editor Report · Decision Letter 5]

12 Nov 2024

PONE-D-24-00348R5Assessing the Drivers of Sexual Behavior and its social determinants in Nepali YouthsPLOS ONE

Dear Dr. Kakchapati,

Thank you for submitting your manuscript to PLOS ONE. After careful consideration, we feel that it has merit but does not fully meet PLOS ONE’s publication criteria as it currently stands. Therefore, we invite you to submit a revised version of the manuscript that addresses the points raised during the review process.

**ACADEMIC EDITOR: **

Dear Dr Kakchapati,

I hope you are doing well.

I would like to kindly remind you to address all of my previous comments and suggestions in the manuscript. In addition, thorough proofreading is essential to ensure the quality and clarity of the work. Please make sure to carefully review the document for any errors.

Once revisions have been made, I kindly ask you to submit two versions of the manuscript: (1) A track-changes version to highlight the changes made. and (2) A clean, revised version of the manuscript without any tracked changes.

Thank you for your attention to these details. I look forward to receiving the updated manuscript.

Kind regards

Shalik Ram Dhital, PhD

Academic Editor

We look forward to receiving your revised manuscript.

Kind regards,

Shalik Ram Dhital, PhD

Academic Editor

PLOS ONE

Journal Requirements:

Additional Editor Comments:

Dear Dr Kakchapati,

I hope you are doing well.

I would like to kindly remind you to address all of my previous comments and suggestions in the manuscript. In addition, thorough proofreading is essential to ensure the quality and clarity of the work. Please make sure to carefully review the document for any errors.

Once revisions have been made, I kindly ask you to submit two versions of the manuscript: (1) A track-changes version to highlight the changes made. and (2) A clean, revised version of the manuscript without any tracked changes.

Thank you for your attention to these details. I look forward to receiving the updated manuscript.

Kind regards

Shalik Ram Dhital, PhD

Academic Editor

---

## [Author Response · Author response to Decision Letter 5]

22 Nov 2024

To,

The Editor,

PLOS ONE

Subject: Response to the query from reviewers

Dear Editor,

Thank you for the evaluation from the reviewers on manuscript entitled on “Assessing the drivers of sexual behavior among youth and its social determinants in Nepal”.

We have responded all of the queries and comments of the reviewers. Please see below for our response to each of the comments/queries. We have also shared the revised manuscript in track changes for your review and approval. 

Thank you very much for reviewing our study. If you need further clarification on this, please do not hesitate to contact us.

Yours sincerely,

Dr. Sampurna Kakchapati

Response to reviewer 

Please keep each author’s ORCID number if available, and Corresponding author’s ORCID Number is mandatory 

Author Response: We had added the ORCID number as suggested. It is mention after the acknowledgement section.

Please add one or two examples of psychological factors 

Author Response: We had examples of psychological factors as suggested in revised manuscript. 

Please check with NDHS data set to verify and make sure these two numbers must be corrected 

Author Response: Yes, we check it and the NDHS datasets are corrected.

Please keep classification of family members in brackets consistent with Table below. 

Author Response: We had added the classification of family members in brackets as suggested.

Please specify other terai caste.

Author Response: We had specified other terai caste as suggested

How did you get this number? Can you please mention it in sample size subsection in Method chapter too?

Author Response: The number include male and female youths who were unmarried. We had added this information in methods section.

---

## [Editor Report · Decision Letter 6]

27 Nov 2024

Assessing the drivers of sexual behavior among youth and its social determinants in Nepal

PONE-D-24-00348R6

Dear Dr. Sampurna

Thank you for your revised paper and for addressing  the previous queries. 

We’re pleased to inform you that your manuscript has been judged scientifically suitable for publication and will be formally accepted for publication once it meets all outstanding technical requirements.

Kind regards,

Shalik Ram Dhital, PhD

Academic Editor

PLOS ONE

---

## [Editor Report · Acceptance letter]

20 Dec 2024

PONE-D-24-00348R6 

PLOS ONE

Dear Dr. Kakchapati, 

I'm pleased to inform you that your manuscript has been deemed suitable for publication in PLOS ONE. Congratulations! Your manuscript is now being handed over to our production team.

Kind regards, 

on behalf of

Dr. Shalik Ram Dhital 

Academic Editor

PLOS ONE